# Non-Asymptotic Pure Exploration by Solving Games

**Rémy Degenne**
Centrum Wiskunde & Informatica
Science Park 123, 1098 XG Amsterdam
`remy.degenne@cwi.nl`

**Wouter M. Koolen**
Centrum Wiskunde & Informatica
Science Park 123, 1098 XG Amsterdam
`wmkoolen@cwi.nl`

**Pierre Ménard**
Inria Lille
40 Avenue Halley, 59650 Villeneuve-d'Ascq
`menardprr@gmail.com`

## Abstract

Pure exploration (aka active testing) is the fundamental task of sequentially gathering information to answer a query about a stochastic environment. Good algorithms make few mistakes and take few samples.

Lower bounds (for multi-armed bandit models with arms in an exponential family) reveal that the sample complexity is determined by the solution to an optimisation problem. The existing state of the art algorithms achieve asymptotic optimality by solving a plug-in estimate of that optimisation problem at each step.

We interpret the optimisation problem as an unknown game, and propose sampling rules based on iterative strategies to estimate and converge to its saddle point. We apply no-regret learners to obtain the first finite confidence guarantees that are adapted to the exponential family and which apply to any pure exploration query and bandit structure. Moreover, our algorithms only use a best response oracle instead of fully solving the optimisation problem.

## 1 Introduction

We study fundamental trade-offs arising in sequential interactive learning. We adopt the framework of Pure Exploration, in which the learning system interacts with its environment by performing a sequence of experiments, with the goal of maximising information gain. We aim to design general, efficient systems that can answer a given query with few experiments yet few mistakes.

As usual, we model the environment by a multi-armed bandit model with exponential family arms, and work in the fixed confidence ($\delta$-PAC) setting. Information-theoretic lower bounds [13] show that a certain number of samples is unavoidable to reach a certain confidence. Moreover, algorithms are developed [13] that match these lower bounds asymptotically, in the small confidence $\delta \to 0$ regime.

Our contribution is a framework for obtaining efficient algorithms with *non-asymptotic guarantees*. The main object of study is the "Pure Exploration Game" [9], a two-player zero-sum game that is central to lower bounds as well as to the widely used GLRT-based stopping rules. We develop iterative methods that provably converge to saddle-point behaviour. The game itself is not known to the learner, and has to be explored and estimated on the fly. Our methods are based on pairs of low-regret algorithms, combined with optimism and tracking. We prove sample complexity guarantees for several combinations of algorithms, and discuss their computational and statistical trade-offs.

The rest of the introduction provides more detail on pure exploration problems, the pure exploration game, the connection between them, and expands on our contribution. We also review related work.

Our model for the environment is a $K$-armed bandit, i.e. distributions $(\nu_1, \ldots, \nu_K)$ on $\mathbb{R}$. We assume throughout that these distributions come from a one-dimensional exponential family, and we denote by $d(\mu, \lambda)$ the relative entropy (Kullback-Leibler divergence) from the distribution with mean $\mu$ to that with mean $\lambda$. A pure exploration problem is parameterised by a set $\mathcal{M}$ of $K$-armed bandit models (the possible environments), a finite set $\mathcal{I}$ of candidate answers and a correct-answer function $i^* : \mathcal{M} \to \mathcal{I}$. We focus on *Best Arm Identification*, for which $i^*(\boldsymbol{\mu}) = \operatorname{argmax}_i \mu_i$ and the *Minimum Threshold* problem, which is defined for any fixed threshold $\gamma$ by $i^*(\boldsymbol{\mu}) = \mathbf{1}_{\{\min_i \mu_i < \gamma\}}$. The goal of the learner is to learn $i^*(\boldsymbol{\mu})$ confidently and efficiently by means of sequentially sampling from the arms of $\boldsymbol{\mu}$, no matter which $\boldsymbol{\mu} \in \mathcal{M}$ it faces. When an algorithm sequentially interacts with $\boldsymbol{\mu}$, we denote by $N_t^k$ and $\hat{\mu}_t^k$ the sample count and empirical mean estimate (these form a sufficient statistic) for each arm $k$ after $t$ rounds. We write $\tau_\delta$ for the time at which the algorithm stops and $\hat{\imath}$ for the answer it recommends. The algorithm is correct (on a particular run) if it recommends $\hat{\imath} = i^*(\boldsymbol{\mu})$ the correct answer for $\boldsymbol{\mu}$. An algorithm is $\delta$-PAC (or $\delta$-correct) if $\mathbb{P}_{\boldsymbol{\mu}}(\hat{\imath} \neq i^*(\boldsymbol{\mu})) \leq \delta$ for each $\boldsymbol{\mu} \in \mathcal{M}$. Among $\delta$-PAC algorithms, we are interested in those minimising the sample complexity $\mathbb{E}_{\boldsymbol{\mu}}[\tau_\delta]$. As it turns out, what can be achieved, and how, is captured by a certain game.

For each $\boldsymbol{\mu} \in \mathcal{M}$, [9] define the two-player zero-sum simultaneous-move *Pure Exploration Game*: MAX plays an arm $k \in [K]$, MIN plays an "alternative" bandit model $\boldsymbol{\lambda} \in \mathcal{M}$ with a different correct answer $i^*(\boldsymbol{\lambda}) \neq i^*(\boldsymbol{\mu})$. We denote the set of such alternatives to answer $i$ by $\neg i = \{\boldsymbol{\lambda} \in \mathcal{M} : i^*(\boldsymbol{\lambda}) \neq i\}$. MAX then receives payoff $d(\mu^k, \lambda^k)$ from MIN. As the payoff is neither concave in $k$ (since discrete) nor convex in $\boldsymbol{\lambda}$ (both domain and divergence are problematic), we will analyse the game by sequencing the moves and considering a mixed strategy for the player moving first. With MAX moving first and playing a mixed strategy $k \sim \boldsymbol{w} \in \triangle_K$ (we identify distributions over $[K]$ and the simplex $\triangle_K$), the value of the game is

$$D_{\boldsymbol{\mu}} := \sup_{\boldsymbol{w} \in \triangle_K} D_{\boldsymbol{\mu}}(\boldsymbol{w}) \qquad \text{where} \qquad D_{\boldsymbol{\mu}}(\boldsymbol{w}) := \inf_{\boldsymbol{\lambda} \in \mathcal{M}: i^*(\boldsymbol{\lambda}) \neq i^*(\boldsymbol{\mu})} \sum_{k=1}^{K} w^k d(\mu^k, \lambda^k). \quad (1)$$

We denote a minimiser of $D_{\boldsymbol{\mu}}$ by $\boldsymbol{w}^*(\boldsymbol{\mu})$ and call it an *oracle allocation*. The analogue where MIN plays first using a mixed strategy $\boldsymbol{\lambda} \sim \boldsymbol{q} \in \mathbb{P}(\neg i^*(\boldsymbol{\mu}))$ (distributions over that set) is proposed and analysed in [9]. Despite the baroque domain of $\boldsymbol{\lambda}$ in (1), there always exist minimax $\boldsymbol{q}$ supported on $\leq K$ points due to dimension constraints.

The Pure Exploration Game is essential to both characterising the complexity of learning, and also to algorithm design. Namely, first, any $\delta$-correct algorithm has sample complexity for each bandit $\boldsymbol{\mu} \in \mathcal{M}$ at least $\mathbb{E}_{\boldsymbol{\mu}}[\tau_\delta] \geq \mathrm{kl}(\delta, 1 - \delta)/D_{\boldsymbol{\mu}} \approx \ln \frac{1}{\delta}/D_{\boldsymbol{\mu}}$, and matching this rate requires sampling proportions $\mathbb{E}_{\boldsymbol{\mu}}[\boldsymbol{N}_{\tau_\delta}]/\mathbb{E}_{\boldsymbol{\mu}}[\tau_\delta]$ converging to $\boldsymbol{w}^*(\boldsymbol{\mu})$ as $\delta \to 0$ [see 13]. Moreover, second, the general approach for obtaining $\delta$-correct algorithms is based on the Generalised Likelihood Ratio Test (GLRT) statistic $Z_t := t D_{\hat{\boldsymbol{\mu}}_t}(\boldsymbol{N}_t/t)$. There are universal thresholds $\beta(t, \delta) \approx \ln \frac{1}{\delta} + \frac{K}{2} \ln \ln \frac{t}{\delta}$ [see e.g. 12, 13, 19, 23] such that $\mathbb{P}_{\boldsymbol{\mu}} \{\exists t : Z_t \geq \beta(\delta, t)\} \leq \delta$ for any $\boldsymbol{\mu} \in \mathcal{M}$. Hence stopping when $Z_t \geq \beta(t, \delta)$ and recommending $\hat{\imath} = i^*(\hat{\boldsymbol{\mu}}_t)$ is $\delta$-correct for any sampling rule. Maximising the GLRT to stop as early as possible is achieved by the sampling proportions $\boldsymbol{N}_t/t = \boldsymbol{w}^*(\hat{\boldsymbol{\mu}}_t)$.

These considerations show that any successful Pure Exploration agent needs to (approximately) solve the Pure Exploration Game $D_{\boldsymbol{\mu}}$. The Track-and-Stop approach, pioneered by [13], ensures that $\hat{\boldsymbol{\mu}}_t \to \boldsymbol{\mu}$ using *forced exploration*, and $\boldsymbol{N}_t/t \to \boldsymbol{w}^*(\hat{\boldsymbol{\mu}}_t)$ using *tracking*. Continuity of $\boldsymbol{w}^*$ and $D_{\boldsymbol{\mu}}$ then yields that $Z_t \approx t D_{\boldsymbol{\mu}}(\boldsymbol{w}^*(\boldsymbol{\mu})) = t D_{\boldsymbol{\mu}}$. The GLRT stopping rule triggers when $t = \beta(\delta, t)/D_{\boldsymbol{\mu}} \approx \ln \frac{1}{\delta}/D_{\boldsymbol{\mu}}$, meeting the lower bound in the asymptotic regime $\delta \to 0$.

**Our contributions.** We explore methods to solve the Pure Exploration game $D_{\boldsymbol{\mu}}$ associated with the unknown bandit model $\boldsymbol{\mu}$, and discusses their statistical and computational trade-offs. We look at solving the game iteratively, by instantiating a low-regret online learner for each player. In particular for the $k$-player we use a self-tuning instance of Exponentiated Gradient called AdaHedge [8]. The $\boldsymbol{\lambda}$-player needs to play a distribution to deal with non-convexity; we consider Follow the Perturbed Leader as well as an ensemble of Online Gradient Descent experts. We show how a combination of optimistic gradient estimates, concentration of measure arguments and regret guarantees combine to deliver the first non-asymptotic sample complexity guarantees (which retain asymptotic optimality for $\delta \to 0$). The advantage of this approach is that it only requires a best response oracle (1, right) instead of a computationally more costly max-min oracle (1, left) employed by Track-and-Stop. Going the other extreme, we also develop Optimistic Track-and-Stop based on a max-max-min oracle (the outer

max implementing optimism over a confidence region for $\boldsymbol{\mu}$), which trades increased computation for tighter sample complexity guarantees with simpler proofs.

Our cocktail sheds new light on the trade-offs involved in the design of pure exploration algorithms. We show how "big-hammer" forced exploration can be refined using problem-adapted optimism. We show how tracking is unnecessary when the $k$ player goes second. We show how computational complexity can be traded off using oracles of various sophistication. And finally, we validate our approach empirically in benchmark experiments at practical $\delta$, and find that our algorithms are either competitive with Track-and-Stop (dense $\boldsymbol{w}^*$) or dominate it (sparse $\boldsymbol{w}^*$).

**Related work** Besides maximising information gain, there is a vast literature on maximising reward in multi-armed bandit models for which a good starting point is [21]. The canonical Pure Exploration problem is Best Arm Identification [10, 3], which is actively studied in the fixed confidence, fixed budget and simple regret settings [21, Ch. 33]. Its sample complexity as a function of the confidence level $\delta$ has been analysed very thoroughly in the (sub)-Gaussian case, where we have a rather complete picture, even including lower order terms [5]. [18] initiated the quest for correct instance-dependent constants for arms from any exponential family. [26] stresses the importance of the "moderate confidence" regime $\delta \gg 0$. Although it is not the focus here, we do believe that it is crucial to obtain the right problem dependence not only in $\ln \frac{1}{\delta}$ but also in $K$ and other structural parameters, as the latter may in practice dominate the sample complexity.

Pure Exploration queries beyond Best Arm include Top-$M$ [15], Thresholding [22], Minimum Threshold [20], Combinatorial Bandits [6], pure-strategy Nash equilibria [29] and Monte-Carlo Tree Search [27]. There is also significant interest in these problems in structured bandit models, including Rank-one [17], Lipschitz [23], Monotonic [14], Unimodal [7] and Unit-Sum [26]. Our framework applies to all these cases. Problems with multiple correct answers were recently considered by [9]. Existing learning strategies do not work unmodified; some fail and others need to be generalised.

Optimism is ubiquitous in bandit optimisation since [1], and was adapted to pure exploration by [16]. We are not aware of optimism being used to solve unknown min-max problems. Optimism was employed in the UCB Frank-Wolfe method by [2] for maximising an unknown smooth function faster. We do not currently know how to make use of such fast rate results. For games the best response value is a non-smooth function of the action.

Using a pair of independent no-regret learners to solve a fixed and known game goes back to [11]. More recently game dynamics were used to explain (Nesterov) acceleration in offline optimisation [28]. Ensuring faster convergence with coordinating learners is an active area of research [25]. Unfortunately, we currently do not know how to obtain an advantage in this way, as our main learning overhead comes from concentration, not regret.

## 2 Algorithms with finite confidence sample complexity bounds

We introduce a family of algorithms, presented as Algorithm 1, with sample complexity bounds for non-asymptotic confidence $\delta$. It uses the following ingredients: the GLRT stopping rule, a saddle point algorithm (possibly formed by two regret minimization algorithms) and optimistic loss estimates.

### 2.1 Model and assumption: sub-Gaussian exponential families.

We suppose that the arm distributions belong to a known one-parameter exponential family. That is, there is a reference measure $\nu_0$ and parameters $\eta_1, \ldots, \eta_K \in \mathbb{R}$ such that the distribution of arm $k \in [K]$ is defined by $d\nu_k/d\nu_0(x) \propto e^{\eta_k x}$. Examples include Gaussians with a given variance or Bernoulli with means in $(0, 1)$. All results can be extended to arms each in a possibly different known exponential family. Let $\Theta$ be the open interval of possible means of such distributions. A distribution $\nu$ is said to be $\sigma^2$-sub-Gaussian if for all $u \in \mathbb{R}$, $\log \mathbb{E}_{X \sim \nu} e^{u(X - \mathbb{E}_{X \sim \nu}[X])} \leq \frac{\sigma^2}{2} u^2$. An exponential family has all distributions sub-Gaussian with constant $\sigma^2$ iff for all $\mu, \lambda \in \Theta$, it verifies $d(\mu, \lambda) \geq \frac{1}{2\sigma^2}(\mu - \lambda)^2$.

**Assumption 1.** The arm distributions belong to sub-Gaussian exponential families with constant $\sigma^2$.

**Assumption 2.** There exists a closed interval $[\mu_{\min}, \mu_{\max}] \subset \Theta$ such that $\mathcal{M} \subseteq [\mu_{\min}, \mu_{\max}]^K$.

---
**Algorithm 1** Pure exploration meta-algorithm.
---
**Require:** Algorithms $\mathcal{A}^k$ and $\mathcal{A}^{\lambda}$, stopping threshold $\beta(t,\delta)$ and exploration bonus $f(t)$.
 1: Sample each arm once and form estimate $\hat{\boldsymbol{\mu}}_K$.
 2: **for** $t = K+1, \dots$ **do**
 3: &emsp;For $k \in [K]$, let $[\alpha_t^k, \beta_t^k] = \{\xi : N_{t-1}^k d(\hat{\mu}_{t-1}^k, \xi) \le f(t-1)\}$. &emsp;&emsp;$\triangleright$ KL confidence intervals
 4: &emsp;Let $\tilde{\boldsymbol{\mu}}_{t-1} = \operatorname{argmin}_{\boldsymbol{\lambda} \in \mathcal{M} \cap \times_{k=1}^K [\alpha_t^k, \beta_t^k]} \sum_{k=1}^K N_{t-1}^k d(\hat{\mu}_{t-1}^k, \lambda^k)$. &emsp;$\triangleright = \hat{\boldsymbol{\mu}}_{t-1}$ if $\hat{\boldsymbol{\mu}}_{t-1} \in \mathcal{M}$
 5: &emsp;Let $i_t = i^*(\tilde{\boldsymbol{\mu}}_{t-1})$.
 6: &emsp;Stop and output $\hat{\imath} = i_t$ **if** $\inf_{\boldsymbol{\lambda} \in \neg i_t} \sum_k N_{t-1}^k d(\hat{\mu}_{t-1}^k, \lambda^k) > \beta(t,\delta)$. &emsp;$\triangleright$ GLRT Stopping rule
 7: &emsp;Get $\boldsymbol{w}_t$ and $\boldsymbol{q}_t$ from $\mathcal{A}_{i_t}^k$ and $\mathcal{A}_{i_t}^{\lambda}$.
 8: &emsp;For $k \in [K]$, let $U_t^k = \max \left\{ f(t-1)/N_{t-1}^k, \max_{\xi \in \{\alpha_t^k, \beta_t^k\}} \mathbb{E}_{\boldsymbol{\lambda} \sim \boldsymbol{q}_t} d(\xi, \lambda^k) \right\}$. &emsp;$\triangleright$ Optimism
 9: &emsp;Feed $\mathcal{A}_{i_t}^k$ the loss $\ell_t^{\boldsymbol{w}}(\boldsymbol{w}) = -\sum_{k=1}^K w^k U_t^k$.
 10: &emsp;Feed $\mathcal{A}_{i_t}^{\lambda}$ the loss $\ell_t^{\boldsymbol{\lambda}}(\boldsymbol{q}) = \mathbb{E}_{\boldsymbol{\lambda} \sim \boldsymbol{q}} \sum_{k=1}^K w_t^k d(\hat{\mu}_{t-1}^k, \lambda^k)$.
 11: &emsp;Pick arm $k_t = \operatorname{argmin}_k N_{t-1}^k / \sum_{s=1}^t w_s^k$. &emsp;&emsp;&emsp;&emsp;$\triangleright$ Cumulative tracking
 12: &emsp;Observe sample $X_t \sim \nu_{k_t}$. Update $\hat{\boldsymbol{\mu}}_t$.
 13: **end for**
---

As a consequence of Assumption 2, there exists $L, D > 0$ such that for all $y \in [\mu_{\min}, \mu_{\max}]$, the function $x \mapsto d(x, y)$ is $L$-Lipschitz on $[\mu_{\min}, \mu_{\max}]$ and $d(x, y) \le D$. Assumption 1 is implied by Assumption 2. Both are discussed in Appendix F. In particular, Assumption 2 can often be relaxed. $L$ and $D$ will appear in the sample complexity bounds but none of our algorithms use them explicitly.

Everywhere below, $\hat{\boldsymbol{\mu}}_t$ denotes the orthogonal projection of the empirical mean onto $[\mu_{\min}, \mu_{\max}]^K$, with one possible exception: the GLRT stopping rule may use it either projected or not, indifferently.

## 2.2 Algorithmic ingredients

**Stopping and recommendation rules.** The algorithm stops if any one of $|\mathcal{I}|$ many GLRT tests succeeds [13]. Let $\mathcal{L}_{\boldsymbol{\mu}}$ denote the likelihood under the model parametrized by $\boldsymbol{\mu}$. The generalized log-likelihood ratio between a set $\Lambda$ and the whole parameter space $\Theta^K$ is

$$\text{GLR}_t^{\Theta^K}(\Lambda) = \log \frac{\sup_{\tilde{\boldsymbol{\mu}} \in \Theta^K} \mathcal{L}_{\tilde{\boldsymbol{\mu}}}(X_1, \dots, X_t)}{\sup_{\boldsymbol{\lambda} \in \Lambda} \mathcal{L}_{\boldsymbol{\lambda}}(X_1, \dots, X_t)} = \inf_{\boldsymbol{\lambda} \in \Lambda} \sum_{k \in [K]} N_t^k d(\hat{\mu}_t^k, \lambda^k) .$$

By concentration of measure arguments, we may find $\beta(t, \delta)$ such that with probability greater than $1 - \delta$, for all $t \in \mathbb{N}$, $\text{GLR}_t^{\Theta^K}(\{\boldsymbol{\mu}\}) \le \beta(t, \delta)$ [see 12, 13, 19, 23]. Test $i \in \mathcal{I}$ succeeds if $\text{GLR}_t^{\Theta^K}(\neg i) > \beta(t, \delta)$. If the algorithm stops because of test $i$, it recommends $\hat{\imath} = i$ (if several tests succeed at the same time, it chooses arbitrarily among these).

**Theorem 1.** *Any algorithm using the GLRT stopping and recommendation rules with threshold $\beta(t, \delta)$ such that $\mathbb{P}_{\boldsymbol{\mu}}\{GLR_t^{\Theta^K}(\{\boldsymbol{\mu}\}) > \beta(t, \delta)\} \le \delta$ is $\delta$-correct.*

**A game with two players** An algorithm is unable to stop at time $t$ if the stopping condition is not met, i.e.

$$\beta(t, \delta) \ge \inf_{\boldsymbol{\lambda} \in \neg i^*(\hat{\boldsymbol{\mu}}_t)} \sum_{k \in [K]} N_t^k d(\hat{\mu}_t^k, \lambda^k) .$$

In order to stop early, the right hand side has to be maximized, i.e. made close to $t \sup_{\boldsymbol{w} \in \triangle_K} \inf_{\boldsymbol{\lambda} \in \neg i^*(\hat{\boldsymbol{\mu}}_t)} \sum_{k \in [K]} w_t^k d(\hat{\mu}_t^k, \lambda^k) = t D_{\hat{\boldsymbol{\mu}}_t} \approx t D_{\boldsymbol{\mu}}$. Then with $\beta(t, \delta) \approx \log 1/\delta + o(t)$ we obtain $t \le \log(1/\delta)/D_{\boldsymbol{\mu}}$ up to lower order terms, i.e. the stopping time is close to optimality.

We propose to approach that max-min saddle-point by implementing two iterative algorithms, $\mathcal{A}^k$ and $\mathcal{A}^{\lambda}$, for the $k$-player and a $\boldsymbol{\lambda}$-player. Our sample complexity bound is a function of two quantities $R_t^k$ and $R_t^{\lambda}$, regret bounds of algorithms $\mathcal{A}^k$ and $\mathcal{A}^{\lambda}$ when used for $t$ steps on appropriate losses.

One player of our choice goes first. The second player can see the action of the first, see the corresponding loss function and use an algorithm with zero regret (e.g. Best-Response or Be-The-Leader). One of the players has to play distributions on its action set. We have one of the following:

1. $\boldsymbol{\lambda}$-player plays first and uses a distribution in $\mathbb{P}(\neg i_t)$. The $k$-player plays $k_t \in [K]$.
2. $k$-player plays first and uses $\boldsymbol{w}_t \in \triangle_K$ (distribution over $[K]$). The $\boldsymbol{\lambda}$-player plays $\boldsymbol{\lambda}_t \in \neg i_t$.
3. Both players play distributions and go in any order, or concurrently.

Algorithm 1 presents two players playing concurrently but can be modified: if for example $\boldsymbol{\lambda}$ plays second, then it gets to see $\ell_t^{\boldsymbol{\lambda}}(\boldsymbol{q})$ before computing $\boldsymbol{q}_t$.

The sampling rule at stage $t$ first computes the most likely answer $i_t$ for $\hat{\boldsymbol{\mu}}_{t-1}$. If the set over which the algorithm optimizes at line 4 is empty, $i_t$ is arbitrary. The $k$-player plays $\boldsymbol{w}_t$ coming from $\mathcal{A}_{i_t}^k$, an instance of $\mathcal{A}^k$ running only on the rounds on which the selected answer is that $i_t$. The $\boldsymbol{\lambda}$-player similarly uses an instance $\mathcal{A}_{i_t}^{\boldsymbol{\lambda}}$ of $\mathcal{A}^{\boldsymbol{\lambda}}$.

**Tracking.** Since a single arm has to be pulled, if the $k$-player plays $\boldsymbol{w} \in \triangle_K$ an additional procedure is needed to translate that play into a sampling rule. We use a so-called tracking procedure, $k_t = \operatorname{argmin}_{k \in [K]} N_{t-1}^k / \sum_{s=1}^t w_s^k$, which ensures that $\sum_{s=1}^t w_s^k - (K-1) \leq N_t^k \leq \sum_{s=1}^t w_s^k$.

**Optimism in face of uncertainty.** Existing algorithms for general pure exploration use forced exploration to ensure convergence of $\hat{\boldsymbol{\mu}}_t$ to $\boldsymbol{\mu}$, making sure that every arm is sampled more than e.g. $\sqrt{t}$ times. We replace that method by the "optimism in face of uncertainty" principle, which gives a more adaptive exploration scheme. While that heuristic is widely used in the bandit literature, this work is its first successful implementation for general pure exploration. In Algorithm 1, the $k$-player algorithm gets an optimistic loss depending on $\boldsymbol{w}_t$ and $\boldsymbol{q}_t$. The $\boldsymbol{\lambda}$-player gets a non-optimistic loss.

## 2.3 Proof scheme and sample complexity result

In order to bound the sample complexity, we introduce a sequence of concentration events $\mathcal{E}_t = \{\forall s \leq t, \forall k \in [K],\ d(\hat{\mu}_s^k, \mu^k) \leq \frac{\widehat{W}((1+a)\log(t))}{N_s^k}\}$ for $a > 0$ and $\widehat{W}(x) = x + \log x + 1/2$. It verifies $\sum_{t=3}^{+\infty} \mathbb{P}_{\boldsymbol{\mu}}(\mathcal{E}_t^c) \leq 2eK/a^2$ (see Appendix B for a proof). The concentration intervals used in Algorihtm 1 are a function of $f(t) = \widehat{W}((1+a)(1+b)\log t)$ for $b > 0$.

**Lemma 1.** *Let $\mathcal{E}_t$ be an event and $T_0(\delta) \in \mathbb{N}$ be such that for $t \geq T_0(\delta)$, $\mathcal{E}_t \subseteq \{\tau_\delta \leq t\}$. Then*

$$\mathbb{E}_{\boldsymbol{\mu}}[\tau_\delta] = \sum_{t=1}^{+\infty} \mathbb{P}\{\tau_\delta > t\} \leq T_0(\delta) + \sum_{t=T_0(\delta)}^{+\infty} \mathbb{P}_{\boldsymbol{\mu}}(\mathcal{E}_t^c).$$

We now present briefly the steps of the proof for the stopping time upper bound before stating our main theorem on the sample complexity of Algorithm 1. These steps are inexact and should be regarded as a guideline and not as rigorous computations. A full proof of our results can be found in the appendices (Appendix B for concentration results, C for tracking and D for the main sample complexity proof). We simplify the presentation by supposing that $i_t = i^*(\boldsymbol{\mu})$ throughout (the main proof will show this may fail only $o(t)$ rounds). For $t < \tau_\delta$, under concentration event $\mathcal{E}_t$,

$$\beta(t,\delta) \geq \inf_{\boldsymbol{\lambda} \in \neg i^*(\boldsymbol{\mu})} \sum_{k \in [K]} N_t^k d(\hat{\mu}_t^k, \lambda^k) \qquad \text{(stopping condition)}$$

$$\geq \inf_{\boldsymbol{\lambda} \in \neg i^*(\boldsymbol{\mu})} \sum_{s \in [t]} \sum_{k \in [K]} w_s^k d(\hat{\mu}_t^k, \lambda^k) - KD \qquad \text{(tracking)}$$

$$\geq \inf_{\boldsymbol{\lambda} \in \neg i^*(\boldsymbol{\mu})} \sum_{s \in [t]} \sum_{k \in [K]} w_s^k d(\hat{\mu}_{s-1}^k, \lambda^k) - \mathcal{O}(\sqrt{t \log(t)}). \qquad \text{(concentration)}$$

The first term is now the infimum of a sum of losses, $\inf_{\boldsymbol{\lambda} \in \neg i^*(\boldsymbol{\mu})} \sum_{s \in [t]} \ell_s^{\boldsymbol{\lambda}}(\boldsymbol{\lambda})$. We use the regret property of the $\boldsymbol{\lambda}$-player's algorithm on those losses, then we introduce optimistic values $U_s^k$ such

that for $\xi^k \in \{\mu^k, \hat{\mu}_{s-1}^k\}$ we have $\mathbb{E}_{\boldsymbol{\lambda} \sim \boldsymbol{q}_s} d(\xi^k, \lambda^k) \leq U_s^k \leq \mathbb{E}_{\boldsymbol{\lambda} \sim \boldsymbol{q}_s} d(\xi^k, \lambda^k) + \mathcal{O}(\sqrt{1/s})$.

$$\inf_{\boldsymbol{\lambda} \in \neg i^*(\boldsymbol{\mu})} \sum_{s \in [t]} \sum_{k \in [K]} w_s^k d(\hat{\mu}_{s-1}^k, \lambda^k) \geq \sum_{s \in [t]} \mathbb{E}_{\boldsymbol{\lambda} \sim \boldsymbol{q}_s} \sum_{k \in [K]} w_s^k d(\hat{\mu}_{s-1}^k, \lambda^k) - R_t^{\boldsymbol{\lambda}} \qquad \text{(regret } \boldsymbol{\lambda}\text{)}$$

$$\geq \sum_{s \in [t]} \sum_{k \in [K]} w_s^k U_s^k - \mathcal{O}(\sqrt{t}) - R_t^{\boldsymbol{\lambda}} \qquad \text{(optimism)}$$

$$\geq \max_{k \in [K]} \sum_{s \in [t]} U_s^k - R_t^k - \mathcal{O}(\sqrt{t}) - R_t^{\boldsymbol{\lambda}} \qquad \text{(regret } \boldsymbol{w}\text{)}$$

$$\geq \max_{k \in [K]} \sum_{s \in [t]} \mathbb{E}_{\boldsymbol{\lambda} \sim \boldsymbol{q}_s} d(\mu^k, \lambda^k) - R_t^k - \mathcal{O}(\sqrt{t}) - R_t^{\boldsymbol{\lambda}} \quad \text{(optimism)}$$

Finally, $1/t \sum_{s \in [t]} \mathbb{E}_{\boldsymbol{\lambda} \sim \boldsymbol{q}_s}$ is itself the expectation of another distribution on $\mathbb{P}(\neg i^*(\boldsymbol{\mu}))$. Hence

$$\max_{k \in [K]} \sum_{s \in [t]} \mathbb{E}_{\boldsymbol{\lambda} \sim \boldsymbol{q}_s} d(\mu^k, \lambda^k) \geq t \inf_{\boldsymbol{q}} \max_k \mathbb{E}_{\boldsymbol{\lambda} \sim \boldsymbol{q}} d(\mu^k, \lambda^k) = t D_{\boldsymbol{\mu}} .$$

Putting these inequalities together, we get finally an inequality on such a $t < \tau_\delta$. The exact result we obtain is the following Theorem, proved in Appendix D.

**Theorem 2.** *Under Assumption 2, the sample complexity of Algorithm 1 on model $\boldsymbol{\mu} \in \mathcal{M}$ is*

$$\mathbb{E}_{\boldsymbol{\mu}}[\tau_\delta] \leq T_0(\delta) + \frac{2eK}{a^2} \quad with \quad T_0(\delta) = \max\{t \in \mathbb{N} : t \leq \frac{\beta(t, \delta)}{D_{\boldsymbol{\mu}}} + C_{\boldsymbol{\mu}}(R_t^{\boldsymbol{\lambda}} + R_t^k + h(t))\} ,$$

*where $C_{\boldsymbol{\mu}}$ depends on $\boldsymbol{\mu}$ and $\mathcal{M}$ and $h(t) = \mathcal{O}(\sqrt{t \log(t)})$. See Appendix D for an exact definition.*

The forms of $h(t)$ and of $T_0(\delta)$ depend on the particular algorithm but we now show how an inequality of that type translates into $T_0(\delta)$. The next lemma is a consequence of the concavity of $t \mapsto \sqrt{t \log t}$.

**Lemma 2.** *Suppose that $t \in \mathbb{R}$ verifies the equation $t - C\sqrt{t \log t} \leq \frac{\log 1/\delta}{D_{\boldsymbol{\mu}}}$. Then for $T_\delta^* = \frac{\log 1/\delta}{D_{\boldsymbol{\mu}}}$,*

$$t \leq \frac{\log 1/\delta}{D_{\boldsymbol{\mu}}} \left( 1 + C \sqrt{\frac{\log T_\delta^*}{T_\delta^*}} \frac{1}{1 - C \frac{1 + \log T_\delta^*}{2\sqrt{T_\delta^* \log T_\delta^*}}} \right) .$$

## 3 Practical Implementations

Next we discuss instantiating no-regret learners. We consider a hierarchy of computational oracles:

1. Min aka Best-Response oracle: obtain for any $i \in \mathcal{I}$, $\boldsymbol{w} \in \triangle_K$ and $\boldsymbol{\xi} \in \Theta^K$ a minimizer in $\neg i$ of $\boldsymbol{\lambda} \mapsto \sum_{k \in [K]} w^k d(\xi^k, \lambda^k)$ .

2. Max-min aka Game-Solving oracle: obtain for any $i \in \mathcal{I}$ and $\boldsymbol{\xi} \in \Theta^K$ a vector $\boldsymbol{w}^* \in \triangle_K$ such that there is a Nash equilibrium $(\boldsymbol{w}^*, \boldsymbol{q}^*) \in \triangle_K \times \mathbb{P}(\neg i)$ for the zero-sum game with reward $d(\xi^k, \lambda^k)$ with the $k$-player using the mixed strategy $\boldsymbol{w}^*$.

3. Max-max-min oracle: for any confidence region $\mathcal{C} = [a_1, b_1] \times \ldots \times [a_K, b_K]$, obtain $(\boldsymbol{\mu}^+, i^+, \boldsymbol{w}^+)$ with $(\boldsymbol{\mu}^+, i^+) = \mathrm{argmax}_{\boldsymbol{\xi} \in \mathcal{C}, i \in \mathcal{I}} \sup_{\boldsymbol{w} \in \triangle_K} \inf_{\boldsymbol{\lambda} \in \neg i} \sum_{k=1}^K w^k d(\xi^k, \lambda^k)$ and $\boldsymbol{w}^+$ a $k$-player strategy of a Nash equilibrium of the game with reward $d(\mu^{+k}, \lambda^k)$.

For Minimum Threshold all oracles can be evaluated in closed form in $O(K)$ time, and the same is true for Best Response in Best Arm Identification. Max-min for Best Arm requires binary search [13] and Max-max-min requires $O(K)$ max-min calls. See [24] for run-time data on Track-and-Stop (max-min oracle) and gradient ascent (min oracle) for Best Arm. Our approach also extends naturally to min-max and max-min-max oracles, which we plan to incorporate in full detail in our future work.

### 3.1 A Learning Algorithm for the $k$-Player vs Best-Response for the $\boldsymbol{\lambda}$-Player

In this section the $k$-player plays first, employing a regret minimization algorithm for linear losses on the simplex to produce $\boldsymbol{w}_t \in \triangle_K$ at time $t$. We pick AdaHedge of [8], which runs in $O(K)$ per round and adapts to the scale of the losses. The $\boldsymbol{\lambda}$-player goes second and can use a zero-regret algorithm: Best-Response. It plays $\boldsymbol{q}_t$ , a Dirac at $\boldsymbol{\lambda}_t \in \mathrm{argmin}_{\boldsymbol{\lambda} \in \neg i_t} \sum_{k \in [K]} w_t^k d(\hat{\mu}_{t-1}^k, \lambda^k)$ .

**Lemma 3.** *AdaHedge has regret* $R_t^k \leq \sqrt{\sum_{s \leq t} b_s^2 \ln K} + \max_{s \leq t} b_s (\frac{4}{3} \ln K + 2)$ *where* $b_s = \max_k U_s^k - \min_k U_s^k \leq \max\{D, f(s)\}$ *is the loss scale in round s, so that* $R_t^k = \mathcal{O}(\sqrt{t \ln K} \ln t)$. *Best-Response has no regret,* $R_t^{\boldsymbol{\lambda}} \leq 0$. *The sample complexity is bounded per Theorem 2.*

We expect that in practice the scale converges to $b_s \to D_{\boldsymbol{\mu}}$ after a transitory startup phase.

**Computational complexity:** one best-response oracle call per time step.

### 3.2  Learning Algorithms for the $\boldsymbol{\lambda}$-Player vs Best Response for the $k$-Player

Using a learner for the $\boldsymbol{\lambda}$-player removes the need for a tracking procedure. In this section the $k$-player goes second and uses Best-Response, with zero regret, i.e. $k_t = \operatorname{argmax}_{k \in [K]} U_t^k$ (see Algorithm 1). After playing $\boldsymbol{q}_t \in \mathbb{P}(\neg i_t)$, the $\boldsymbol{\lambda}$-player suffers loss $\mathbb{E}_{\boldsymbol{\lambda} \sim \boldsymbol{q}_t} d(\hat{\mu}_{t-1}^{k_t}, \lambda^{k_t})$.

Most existing regret minimization algorithms do not apply since the function $\lambda \mapsto d(\mu, \lambda)$ is not convex in general and the action set $\neg i_t$ is also not convex. The challenge is to come up with an algorithm able to play distributions with only access to a best-response oracle.

**Follow-The-Perturbed-Leader.**  Follow-The-Perturbed-Leader can sample points from a distribution on $\mathbb{P}(\neg i)$ by only using best-response oracle calls on $\neg i$. The version we use here incorporates all the information available to the $\boldsymbol{\lambda}$-player: the loss of $\boldsymbol{\lambda} \in \neg i_t$ will be $d(\hat{\mu}_{t-1}^{k_t}, \lambda^{k_t})$ where the only unknown quantity is $k_t$. Let $\sigma_t \in \mathbb{R}^K$ be a random vector with independent exponentially distributed coordinates. The idea is that the distribution $\boldsymbol{q}_t$ played by the $\boldsymbol{\lambda}$-player should be the distribution of

$$\operatorname*{argmin}_{\boldsymbol{\lambda} \in \neg i_t} \sum_{s=1}^{t-1} d(\hat{\mu}_{s-1}^{k_s}, \lambda^{k_s}) + \sum_{k=1}^{K} \sigma_t^k d(\hat{\mu}_{t-1}^k, \lambda^k) \,.$$

We show in Appendix E.2 that this argmin can be computed by a single best-response oracle call. However, the $k$-player has to be able to compute the best response to $\boldsymbol{q}_t$. Since we cannot get the above distribution exactly, we instead take for $\boldsymbol{q}_t$ an empirical distribution from $t$ samples. A regret bound $R_t^{\boldsymbol{\lambda}} = \mathcal{O}(\sqrt{t \log t})$ for that algorithm is in Appendix E.2. The sample complexity is then bounded by Theorem 2.

**Computational complexity:** $t$ best-response oracle calls at time step $t$.

**Online Gradient Descent.**  While the learning problem for $\boldsymbol{\lambda}$ is hard in general, in several common cases the sets $\neg i$ have a simple structure. If these sets are unions of a finite number $J$ of convex sets and $\lambda \mapsto d(\mu, \lambda)$ is convex (i.e. for Gaussian or Bernoulli arm distributions), then we can use off-the-shelf regret algorithms. One gradient descent learner can be used on each convex set, and these $J$ experts are then aggregated by an exponential weights algorithm. This procedure would have $\mathcal{O}(\sqrt{t})$ regret. The computational complexity is $J$ (convex) best-response oracle calls per time step.

### 3.3  Optimistic Track-and-Stop.

At stage $t$, this algorithm computes $(\boldsymbol{\mu}^+, i_t) = \operatorname{argmax}_{\boldsymbol{\xi}, i} \sup_{\boldsymbol{w} \in \triangle_K} \inf_{\boldsymbol{\lambda} \in \neg i} \sum_{k=1}^{K} w^k d(\xi^k, \lambda^k)$ where $\boldsymbol{\xi}$ ranges over all points in $\Theta^K$ in a confidence region around $\hat{\boldsymbol{\mu}}_{t-1}$ and $i \in \mathcal{I}$. Then, the $k$-player plays $\boldsymbol{w}_t$ such that there exists a Nash equilibrium $(\boldsymbol{w}_t, \boldsymbol{q}_t)$ of the game with reward $d(\mu^{+k}, \lambda^k)$. The proof of its sample complexity bound proceeds slightly differently from the sketch of part 2.3, although the ingredients are still the GLRT, concentration, optimism and game-solving. The proof of the following lemma can be found in appendix E.2.

**Lemma 4.** *Take $b = 1$ in the definition of $f(t)$. Let $h(t) = 2\sqrt{t} D_{\boldsymbol{\mu}} + 3L\sqrt{2\sigma^2 f(t)}(K^2 + (2\sqrt{2} + \frac{1}{3})\sqrt{Kt}) + f(t)(K^2 + 2K \log(t/K)) + KD$. Then the expected sample complexity is at most $T_0(\delta) + \frac{2eK}{a^2}$, where $T_0(\delta)$ is the maximal $t \in \mathbb{N}$ such that $t \leq (\beta(t, \delta) + h(t))/D_{\boldsymbol{\mu}}$.*

Note: the $K^2$ factors are due to the tracking. We conjecture that they should be $K \log K$ instead.

**Computational complexity:** one max-max-min oracle call per time step.

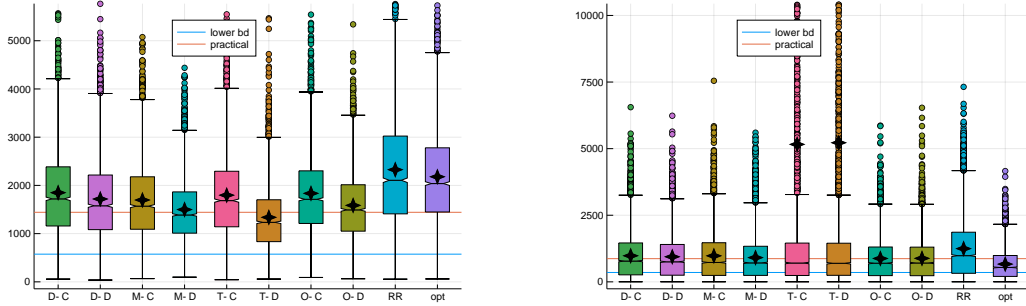

(a) Best Arm for Bernoulli bandit model $\boldsymbol{\mu} = (0.3, 0.21, 0.2, 0.19, 0.18)$. The oracle weights are $\boldsymbol{w}^* = (0.34, 0.25, 0.18, 0.13, 0.10)$.

(b) Minimum Threshold for Gaussian bandit model $\boldsymbol{\mu} = (0.5, 0.6)$ with threshold $\gamma = 0.6$, $\boldsymbol{w}^* = \boldsymbol{e}_1$. Note the excessive sample complexity of T-C/ T-D.

Figure 1: Selected experiments. In both cases $\delta = 0.1$. Plots based on 3000 and 5000 runs.

This algorithm is the most computationally expensive but has the best sample complexity upper bound, has a simpler proof and works well in experiments where computing the max-max-min oracle is feasible, like the Best Arm and Minimum Threshold problems (see section 4).

## 4 Experiments

The goal of our experiments is to empirically validate Algorithm 1 on benchmark problems for practical $\delta$. We use stylised stopping threshold $\beta(\delta, t) = \ln \frac{1 + \ln t}{\delta}$ and exploration bonus $f(t) = \ln t$. Both are unlicensed by theory yet conservative in practise (the error frequency is way below $\delta$). We use the following letter coding to designate sampling rules: **D** for AdaHedge vs Best-Response as advocated in Section 3.1, **T** for Track-and-Stop of [13], **M** for the Gradient Ascent algorithm of [24], **O** for Optimistic Track-and-Stop from Section 3.3, **RR** for uniform, and **opt** for following the oracle proportions $\boldsymbol{w}^*(\boldsymbol{\mu})$. We also ran all our experiments on a simplification of **D** that uses a single learner instead of partitioning the rounds according to $i_t$. We omit it from the results, as it was always within a few percent of **D**. We append **-C** or **-D** to indicate whether cumulative ($\boldsymbol{N}_t \rightsquigarrow \sum_{s \leq t} \boldsymbol{w}_s$) or direct ($\boldsymbol{N}_t \rightsquigarrow t\boldsymbol{w}_t$) tracking [13] is employed. We finally note that we tune the learning rate of **M** in terms of (the unknown) $D_{\boldsymbol{\mu}}$.

We perform two series of experiments, one on Best Arm instances from [13, 24], and one on Minimum Threshold instances from [20]. Two selected experiments are shown in Figure 1, the others are included in Appendix G. We contrast the empirical sample complexity with the lower bound $\mathrm{kl}(\delta, 1 - \delta)/D_{\boldsymbol{\mu}}$, and with a more "practical" version, which indicates the time $t$ for which $t = \beta(t, \delta)/D_{\boldsymbol{\mu}}$, which is, approximately, the first time at which the GLRT stopping rule crosses the threshold $\beta$.

We see in Figures 1(a) and 1(b) that direct tracking **-D** has the advantage over cumulative tracking **-C** across the board, and that uniform sampling **RR** is sub-optimal as expected. In Figure 1(a) we see that **T** performs best, closely followed by **M** and **O**. Sampling from the oracle weights **opt** performs surprisingly poorly (as also observed in [26, Table 1]). The main message of Figure 1(b) is that **T** can be highly sub-optimal. We comment on the reason in Appendix G.2. Asymptotic optimality of **T** implies that this effect disappears as $\delta \to 0$. However, for this example this kicks in excruciatingly slowly. Figure 5 shows that **T** is still not competitive at $\delta = 10^{-20}$. On the other hand, **O** performs best, closely followed by **M** and then **D**. Practically, we recommend using **O** if its computational cost is acceptable, **M** if an estimate of the problem scale is available for tuning, and **D** otherwise.

The gap between **opt** and **T** (or **O**) shows that Track-and-Stop outperforms its design motivation. It is an exciting open problem to understand exactly why, and to optimise for stopping early ($\boldsymbol{N}_t/t \approx \boldsymbol{w}^*(\hat{\boldsymbol{\mu}}_t)$) while ensuring optimality ($\mathbb{E}_{\boldsymbol{\mu}}[\boldsymbol{N}_\tau]/\mathbb{E}_{\boldsymbol{\mu}}[\tau] \approx \boldsymbol{w}^*(\boldsymbol{\mu})$).

# 5  Conclusion

We leveraged the game point of view of the pure exploration problem, together with the use of the optimism principle, to derive algorithms with sample complexity guarantees for non-asymptotic confidence. Varying the flavours of optimism and saddle-point strategies leads to procedures with diverse tradeoffs between sample and computational complexities. Our sample complexity bounds attain asymptotic optimality while offering guarantees for moderate confidence and the obtained algorithms are empirically sound. Our bounds however most probably do not depend optimally on the problem parameters, like the number of arms $K$. For BAI and the Top-K arms problems, lower bounds with lower order terms as well as matching algorithms were derived by [26]. A generalization of such lower bounds to the general pure exploration problem could shed light upon the optimal complexity across the full confidence spectrum.

The richness of existing saddle-point iterative algorithms may bring improved performance over our relatively simple choices. A smart algorithm could possibly take advantage of the stochastic nature of the losses instead of treating them as completely adversarial.

### Acknowledgements

We are grateful to Zakaria Mhammedi and Emilie Kaufmann for multiple generous discussions. Travel funding was provided by INRIA Associate Team [6]PAC. The experiments were carried out on the Dutch national e-infrastructure with the support of SURF Cooperative.

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
