[Supplementary Material]

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

# A   Likelihood Ratio and Exponential Families

## A.1   Canonical one-parameter exponential families

We suppose that all arms have distributions in a canonical one-parameter exponential family. That is, there is a reference measure $\nu_0$ and parameters $\eta_1, \ldots, \eta_K \in \mathbb{R}$ such that the distribution of arm $k \in [K]$ is defined by

$$d\nu_k/d\nu_0(x) \propto e^{\eta_k x - \psi(\eta_k)} \qquad \text{with} \qquad \psi(\eta) = \log \mathbb{E}_{X \sim \nu_0} e^{\eta x} .$$

Let $\phi$ be the convex conjugate of $\psi$, i.e. $\phi(x) = \sup_{y \in \mathrm{dom}\ \psi}(xy - \psi(y))$. Let $\Theta \subset \mathbb{R}$ be the open interval on which the first derivative $\phi'$ is defined. The Kullback-Leibler divergence between the distributions of the exponential family with means $\mu$ and $\lambda$ in $\Theta$ is

$$d(\mu, \lambda) = \phi(\mu) - \phi(\lambda) - (\mu - \lambda)\phi'(\lambda) .$$

A distribution $\nu$ is said to be $\sigma^2$-sub-Gaussian if for all $u \in \mathbb{R}$, $\log \mathbb{E}_{X \sim \nu} e^{u(X - \mathbb{E}_{X \sim \nu}[X])} \leq \frac{\sigma^2}{2} u^2$. A canonical one-parameter exponential family has all distributions sub-Gaussian with constant $\sigma^2$ iff for all $\mu, \lambda \in \Theta$, it verifies $d(\mu, \lambda) \geq \frac{1}{2\sigma^2}(\mu - \lambda)^2$.

## A.2   The Generalized log-likelihood ratio

The generalized log-likelihood ratio between the whole model space $\mathcal{M}$ and a subset $\Lambda \subseteq \mathcal{M}$ is

$$\mathrm{GLR}_t^{\mathcal{M}}(\Lambda) = \log \frac{\sup_{\tilde{\boldsymbol{\mu}} \in \mathcal{M}} \mathcal{L}_{\tilde{\boldsymbol{\mu}}}(X_1, \ldots, X_t)}{\sup_{\boldsymbol{\lambda} \in \Lambda} \mathcal{L}_{\boldsymbol{\lambda}}(X_1, \ldots, X_t)} .$$

In the case of a canonical one-parameter exponential family, the likelihood of the model with means $\boldsymbol{\mu}$ is

$$\mathcal{L}_{\boldsymbol{\mu}}(X_1, \ldots, X_t) = \prod_{s=1}^{t} \exp(\phi'(\mu^{k_s})(X_s - \mu^{k_s}) + \phi(\mu^{k_s}))d\nu_0(X_s)$$

For $\boldsymbol{\xi}, \boldsymbol{\lambda} \in \mathcal{M}$ two mean vectors,

$$\log \frac{\mathcal{L}_{\boldsymbol{\xi}}(X_1, \ldots, X_t)}{\mathcal{L}_{\boldsymbol{\lambda}}(X_1, \ldots, X_t)} = \sum_{s=1}^{t} d(X_s, \lambda^{k_s}) - d(X_s, \xi^{k_s}) = \sum_{k=1}^{K} N_t^k [d(\hat{\mu}_t^k, \lambda^k) - d(\hat{\mu}_t^k, \xi^k)] .$$

The maximum likelihood estimator $\tilde{\boldsymbol{\mu}}_t$ corresponding to the data $X_1, \ldots, X_t$ is

$$\tilde{\boldsymbol{\mu}}_t = \underset{\boldsymbol{\lambda} \in \mathcal{M}}{\mathrm{argmin}} \sum_{k=1}^{K} N_t^k d(\hat{\mu}_t^k, \lambda^k) .$$

The GLR for set $\Lambda$ is

$$\mathrm{GLR}_t^{\mathcal{M}}(\Lambda) = \underset{\boldsymbol{\lambda} \in \Lambda}{\mathrm{argmin}} \sum_{k=1}^{K} N_t^k d(\hat{\mu}_t^k, \lambda^k) - \underset{\boldsymbol{\lambda} \in \mathcal{M}}{\mathrm{argmin}} \sum_{k=1}^{K} N_t^k d(\hat{\mu}_t^k, \lambda^k)$$

$$= \underset{\boldsymbol{\lambda} \in \Lambda}{\mathrm{argmin}} \sum_{k=1}^{K} N_t^k d(\hat{\mu}_t^k, \lambda^k) - \sum_{k=1}^{K} N_t^k d(\hat{\mu}_t^k, \tilde{\mu}_t^k) .$$

# B   Concentration Lemmas

## B.1   Concentration bounds

For $x > 0$, let $\widehat{W}(x) = x + \log(x)$. Let $W_{-1}$ be the negative branch of the Lambert W function and for $x \geq 1$, $\overline{W}(x) = -W_{-1}(-e^{-x})$. Then

- For $x, y \geq 1$, $x - \log x \geq y \Leftrightarrow x \geq \overline{W}(y)$ .

- For $x > 1$, $\widehat{W}(x) \leq \overline{W}(x) \leq \widehat{W}(x) + \min\{\frac{1}{2}, \frac{1}{\sqrt{x}}\}$ .

**Lemma 5** ([12]). *Let $Y_1^k, \ldots, Y_t^k$ be i.i.d. random variables in a canonical one-parameter exponential family with mean $\mu^k$. Then for $\alpha > 0$,*

$$\mathbb{P}_{\boldsymbol{\mu}}\left\{\exists s \leq t, sd(\frac{1}{s}\sum_{r=1}^{s} Y_r^k, \mu^k) \geq \alpha\right\} \leq 2e\log(t)e^{-(\alpha - \log \alpha)} .$$

Remark that for $s \leq t$, the number of pulls verifies $N_s^k \leq t$. For $t > e$ and $\alpha = \overline{W}((1+a)\log t)$ with $a > 0$, the lemma above implies

$$\mathbb{P}_{\boldsymbol{\mu}}\left\{\exists s \leq t, N_s^k d(\hat{\mu}_s^k, \mu^k) \geq \overline{W}((1+a)\log t)\right\} \leq 2e\frac{\log t}{t^{1+a}} .$$

**Definition 1.** Let $f(s) = \overline{W}((1+a)(1+b)\log s)$.

For $s \geq t^{1/(1+b)}$, $\overline{W}((1+a)(1+b)\log s) \geq \overline{W}((1+a)\log t)$, and when the event above happens,

$$d(\hat{\mu}_s^k, \mu^k) \leq \frac{f(s)}{N_s^k} .$$

### B.2  Main concentration event

Concentration event for $t \geq 3$:
$$\mathcal{E}_t = \left\{\forall s \leq t, \forall k \in [K] \; N_s^k d(\hat{\mu}_s^k, \mu^k) \leq \overline{W}((1+a)\log t)\right\}$$

**Lemma 6.**

$$\forall t \geq 3, \; \mathbb{P}_{\boldsymbol{\mu}}(\mathcal{E}_t^c) \leq 2eK\frac{\log t}{t^{1+a}} , \qquad \sum_{t=3}^{+\infty}\mathbb{P}_{\boldsymbol{\mu}}(\mathcal{E}_t^c) \leq \frac{2eK}{a^2} .$$

*Proof.*

$$\sum_{t=3}^{+\infty}\mathbb{P}_{\boldsymbol{\mu}}(\mathcal{E}_t^c) \leq 2eK\sum_{t=3}^{+\infty}\frac{\log t}{t^{1+a}} \leq 2eK\int_{x=1}^{+\infty}\frac{\log x}{x^{1+a}}dx = \frac{2eK}{a^2} .$$

$\square$

## C  Tracking

**Lemma 7.** *Let $(\boldsymbol{w}_s)_{s\in\mathbb{N}} \in \triangle_K^{\mathbb{N}}$ be vectors in the simplex with $\boldsymbol{w}_1, \ldots, \boldsymbol{w}_K$ equal to the basis vectors. We recursively define for $t \in \mathbb{N}$,*

$$\forall k \in [K], \; N_K^k = 1 ,$$

$$\forall t \geq K+1, \; k_t = \underset{k}{\operatorname{argmin}} \frac{N_{t-1}^k}{\sum_{s=1}^{t} w_s^k} , \qquad \forall k \in [K], \; N_t^k = \sum_{s=1}^{t}\mathbb{I}\{k_s = k\} .$$

*The tie-breaking for the* argmin *is arbitrary. Then for all $t \geq K$, all $k \in [K]$,*

$$\sum_{s=1}^{t} w_s^k - (K-1) \leq N_t^k \leq \sum_{s=1}^{t} w_s^k + 1 .$$

*Proof.* Let $\Sigma_t^k = \sum_{s=1}^{t} w_s^k$. We start by proving the inequality on the right by induction. At $t = K$, for all $k$, $N_K^k = \Sigma_K^k = 1$.

Suppose now that $N_s^i \leq \Sigma_s^i + 1$ for all $i \in [K]$ and all $s \leq t-1$. We prove that it also holds for $t$.

If $i \neq k_t$, by the induction hypothesis, $N_{t-1}^i \leq \Sigma_{t-1}^i + 1$. We obtain $N_t^i = N_{t-1}^i \leq \Sigma_{t-1}^i + 1 \leq \Sigma_t^i + 1$.

If $i = k_t$, we use that $\sum_{j=1}^{K} N_{t-1}^j = t - 1$ and $\sum_{j=1}^{K} \Sigma_t^j = t$ to say that $\min_j \frac{N_{t-1}^j}{\Sigma_t^j} \leq \frac{t-1}{t} \leq 1$. Since $k_t$ realizes that minimum, we have

$$\frac{N_t^{k_t}}{\Sigma_t^{k_t}} = \frac{N_{t-1}^{k_t}}{\Sigma_t^{k_t}} + \frac{1}{\Sigma_t^{k_t}} \leq 1 + \frac{1}{\Sigma_t^{k_t}} \ .$$

The inequality is proved for all $k \in [K]$ at $t$.

The lower bound for $N_t^i$ follows from the fact that $\sum_{i=1}^{K} N_t^i = \sum_{i=1}^{K} \Sigma_t^i = t$.

$$N_t^i = t - \sum_{j \neq i} N_t^j \geq t - \sum_{j \neq i} (\Sigma_t^j + 1) = \Sigma_t^i - (K - 1) \ .$$

$\square$

**Lemma 8.** *For $t \geq t_0 \geq 1$ and $(x_s)_{s \in [t]}$ non-negative real numbers such that $\sum_{s=1}^{t_0-1} x_s > 0$,*

$$\sum_{s=t_0}^{t} \frac{x_s}{\sqrt{\sum_{r=1}^{s} x_r}} \leq 2\sqrt{\sum_{s=1}^{t} x_s} - 2\sqrt{\sum_{s=1}^{t_0-1} x_s} \ .$$

$$\sum_{s=t_0}^{t} \frac{x_s}{\sum_{r=1}^{s} x_r} \leq \log(\sum_{s=1}^{t} x_s) - \log(\sum_{s=1}^{t_0-1} x_s) \ .$$

*Proof.* By concavity of $x \mapsto \sqrt{x}$, we have $\sqrt{x} \leq \sqrt{x + y} - \frac{y}{2\sqrt{x+y}}$. We obtain $\frac{x_s}{\sqrt{\sum_{r=1}^{s} x_r}} \leq 2(\sqrt{\sum_{r=1}^{s} x_r} - \sqrt{\sum_{r=1}^{s-1} x_r})$. The sum is then telescopic. The second result uses the concavity of $x \mapsto \log(x)$. $\square$

**Lemma 9.** *Let $(\boldsymbol{w}_s)_{s \in \mathbb{N}} \in \triangle_K^{\mathbb{N}}$ be vectors in the simplex. Let $N_t$ be defined as in Lemma 7. Then*

$$\sum_{k=1}^{K} \sum_{s=K}^{t} \frac{w_s^k}{\sqrt{N_s^k}} \leq K^2 + 2\sqrt{Kt} \quad and \quad \sum_{k=1}^{K} \sum_{s=K+1}^{t} \frac{w_s^k}{\sqrt{N_{s-1}^k}} \leq K^2 + 2\sqrt{2Kt} \ .$$

*Proof.* We first prove the inequality on the left. Let $t_0^k$ be the first time such that $\sum_{r=1}^{t_0^k-1} w_r^k > K - 1$. Then

$$\sum_{s=K}^{t} \frac{w_s^k}{\sqrt{N_s^k}} = \sum_{s=K}^{t_0^k-1} \frac{w_s^k}{\sqrt{N_s^k}} + \sum_{s=t_0^k}^{t} \frac{w_s^k}{\sqrt{N_s^k}} \leq \sum_{s=K}^{t_0^k-1} w_s^k + \sum_{s=t_0^k}^{t} \frac{w_s^k}{\sqrt{N_s^k}} \leq K + \sum_{s=t_0^k}^{t} \frac{w_s^k}{\sqrt{N_s^k}} \ .$$

By the tracking property of Lemma 7,

$$\sum_{s=t_0^k}^{t} \frac{w_s^k}{\sqrt{N_s^k}} \leq \sum_{s=t_0^k}^{t} \frac{w_s^k}{\sqrt{\sum_{r=1}^{s} w_r^k - (K - 1)}} \ .$$

By Lemma 8,

$$\sum_{s=t_0^k}^{t} \frac{w_s^k}{\sqrt{\sum_{r=1}^{s} w_r^k - (K - 1)}} \leq 2\sqrt{\sum_{s=1}^{t} w_s^k - (K - 1)} - 2\sqrt{\sum_{s=1}^{t_0^k} w_s^k - (K - 1)} \leq 2\sqrt{\sum_{s=1}^{t} w_s^k} \ .$$

Putting all these computations together, we obtain

$$\sum_{k=1}^{K} \sum_{s=K}^{t} \frac{w_s^k}{\sqrt{N_s^k}} \leq K^2 + 2\sum_{k=1}^{K} \sqrt{\sum_{s=1}^{t} w_s^k} \leq K^2 + 2\sqrt{Kt} \ .$$

We now prove the inequality on the right. For $s$ such that $N_{s-1}^k \geq 1$, we have $N_{s-1}^k \geq \frac{1}{2} N_s^k$. We remark that this is true for all $s \geq K$, apply it to the sum starting from $t_0^k$, and obtain the wanted inequality. $\square$

# D Sample complexity proof

## D.1 Upper confidence bounds

At stage $t$, we compute the empirical mean vector $\hat{\boldsymbol{\mu}}_{t-1}$ and the mixed strategies of the two players $\boldsymbol{w}_t$ and $\boldsymbol{q}_t$. A concentration event ensures that for all $k \in [K]$, both $\mu^k$ and $\hat{\mu}^k_{t-1}$ belong to an interval $[a^k_t, b^k_t]$. We introduce two types of coordinate-wise upper confidence bounds (UCB). The first type is a vector $U_t \in \mathbb{R}^K$ such that

$$\forall k \in [K], \forall \xi^k \in [a^k_t, b^k_t], \ U^k_t \geq \mathbb{E}_{\boldsymbol{\lambda} \sim \boldsymbol{q}_t} \, d(\xi^k, \lambda^k) \,.$$

The second type is a function of $\boldsymbol{\lambda}$, $U^k_t(\boldsymbol{\lambda})$ such that

$$\forall k \in [K], \forall \boldsymbol{\lambda} \in \mathcal{M}, \forall \xi^k \in [a^k_t, b^k_t], \ U^k_t(\boldsymbol{\lambda}) \geq d(\xi^k, \lambda^k) \,.$$

Let $[\alpha^k_t, \beta^k_t]$ be the intersection of $[\mu_{\min}, \mu_{\max}]$ and the interval $\{\xi \in \Theta \ : \ d(\hat{\mu}^k_{t-1}, \xi) \leq \frac{f(t-1)}{N^k_{t-1}}\}$, where $f$ is defined in Definition 1 in section B.

Let $[a^t_t, b^k_t] = [\mu_{\min}, \mu_{\max}] \cap [\hat{\mu}^k_{t-1} - \sqrt{2\sigma^2 \frac{f(t-1)}{N^k_{t-1}}}, \hat{\mu}^k_{t-1} + \sqrt{2\sigma^2 \frac{f(t-1)}{N^k_{t-1}}}]$.

We consider the following UCBs.

1. $U^{k\,(1)}_t = \max \left\{ \frac{f(t-1)}{N^k_{t-1}}, \max_{\xi \in [\alpha^k_t, \beta^k_t]} \mathbb{E}_{\boldsymbol{\lambda} \sim \boldsymbol{q}_t} \, d(\xi, \lambda^k) \right\}$.

2. $U^{k\,(2)}_t = \max \left\{ \frac{f(t-1)}{N^k_{t-1}}, \max_{\xi \in [a^k_t, b^k_t]} \mathbb{E}_{\boldsymbol{\lambda} \sim \boldsymbol{q}_t} \, d(\xi, \lambda^k) \right\}$.

3. $U^{k\,(1)}_t(\boldsymbol{\lambda}) = \max \left\{ \frac{f(t-1)}{N^k_{t-1}}, \max_{\xi \in [\alpha^k_t, \beta^k_t]} d(\xi, \lambda^k) \right\}$.

4. $U^{k\,(2)}_t(\boldsymbol{\lambda}) = \max \left\{ \frac{f(t-1)}{N^k_{t-1}}, \max_{\xi \in [a^k_t, b^k_t]} d(\xi, \lambda^k) \right\}$.

The UCBs indexed by $(2)$ are larger but potentially easier to compute that the ones indexed by $(1)$, since $a^k_t$ and $b^k_t$ are easier to compute than $\alpha^k_t$ and $\beta^k_t$. The next lemma simplifies the computation of the UCBs.

**Lemma 10.** *In all the UCBs introduced, the maximum over the interval is attained at one of the two extremal points.*

*Proof.* We need to prove that a function of the form $\xi \mapsto \mathbb{E}_{\boldsymbol{\lambda} \sim \boldsymbol{q}} \, d(\xi, \lambda^k)$ attains its maximum at an extremity of any interval. That function has derivative equal to $\phi'(\xi) - \mathbb{E}_{\boldsymbol{\lambda} \sim \boldsymbol{q}} \, \phi'(\lambda^k)$. Since $\phi'$ is increasing, that derivative is negative below a point and positive afterwards. Hence the function is decreasing then increasing. We obtain that its maximum is indeed attained on an extremity of the interval. $\square$

**Lemma 11.** *For all $k \in [K]$, all $t \in \mathbb{N}$, $U^{k\,(1)}_t \leq U^{k\,(2)}_t$. Furthermore for all $\boldsymbol{\lambda} \in \mathcal{M}$, $U^{k\,(1)}_t(\boldsymbol{\lambda}) \leq U^{k\,(2)}_t(\boldsymbol{\lambda})$.*

*Proof.* By the sub-Gaussian assumption 1, $[\alpha^k_t, \beta^k_t] \subseteq [a^k_t, b^k_t]$. $\square$

**Lemma 12.** *$U^{k\,(1)}_t$ and $U^{k\,(2)}_t$ verify $\forall \xi \in [\alpha^k_t, \beta^k_t]$, $U^k_t \geq \mathbb{E}_{\boldsymbol{\lambda} \sim \boldsymbol{q}_t} \, d(\xi, \lambda^k)$. $U^{k\,(1)}_t(\boldsymbol{\lambda})$ and $U^{k\,(2)}_t(\boldsymbol{\lambda})$ verify $\forall \xi \in [\alpha^k_t, \beta^k_t]$, $U^k_t \geq d(\xi, \lambda^k)$.*

*Proof.* It is true for $U^{k\,(1)}_t$ and $U^{k\,(1)}_t(\boldsymbol{\lambda})$ by definition and true for $U^{k\,(2)}_t$ and $U^{k\,(2)}_t(\boldsymbol{\lambda})$ by Lemma 11. $\square$

The analysis will proceed identically with $U^{k\,(1)}_t$ or $U^{k\,(2)}_t$ (resp. $U^{k\,(1)}_t(\boldsymbol{\lambda})$ or $U^{k\,(2)}_t(\boldsymbol{\lambda})$), which will be denoted simply by $U^k_t$ (resp. $U^k_t(\boldsymbol{\lambda})$). The following lemma is an immediate consequence of the definition.

**Lemma 13.** *All UCBs presented verify $U_t^k \geq \frac{f(t-1)}{N_{t-1}^k}$ (resp. $U_t^k(\boldsymbol{\lambda}) \geq \frac{f(t-1)}{N_{t-1}^k}$).*

This lower bound is the reason the UCBs are computed as the maximum of some expression and $\frac{f(t-1)}{N_{t-1}^k}$. But for $U_t^{k(2)}$ and $U_t^{k(2)}(\boldsymbol{\lambda})$, that lower bound is also obtained automatically as soon as $[a_t^k, b_t^k] \subseteq [\mu_{\min}, \mu_{\max}]$. Indeed in that case

$$U_t^{k(2)} \geq \min\{d(\hat{\mu}_{t-1}^k - \sqrt{2\sigma^2 \tfrac{f(t-1)}{N_{t-1}^k}}, \hat{\mu}_{t-1}^k), d(\hat{\mu}_{t-1}^k + \sqrt{2\sigma^2 \tfrac{f(t-1)}{N_{t-1}^k}}, \hat{\mu}_{t-1}^k)\} \ .$$

From the sub-Gaussian assumption, they are both bigger than $\frac{f(t-1)}{N_{t-1}^k}$.

## D.2 Saddle point algorithms

Let $\Lambda$ be a subset of $\mathcal{M}$.

**Definition 2.** In the context of this proof, an algorithm playing sequences $(\boldsymbol{w}_s, \boldsymbol{q}_s)_{s \leq t} \in (\triangle_K \times \mathbb{P}(\Lambda))^{[t]}$ is said to be an approximate optimistic saddle point algorithm with slack $x_t$ if

$$\inf_{\boldsymbol{\lambda} \in \Lambda} \sum_{s=1}^t \sum_{k=1}^K w_s^k d(\hat{\mu}_{s-1}^k, \lambda^k) \geq \max_k \sum_{s=1}^t \mathbb{E}_{\boldsymbol{\lambda} \sim \boldsymbol{q}_s} U_s^k(\boldsymbol{\lambda}) - x_t \ ,$$

$$\text{or} \quad \inf_{\boldsymbol{\lambda} \in \Lambda} \sum_{s=1}^t \sum_{k=1}^K w_s^k d(\hat{\mu}_{s-1}^k, \lambda^k) \geq \max_k \sum_{s=1}^t U_s^k - x_t \ .$$

We now show two ways to prove that a procedure is an approximate optimistic saddle point algorithm, introducing either upper bounds $U_t^k(\boldsymbol{\lambda})$ or $U_t^k$.

**Introduce UCBs, then use a saddle point property.** We can start by replacing $d(\hat{\mu}_{s-1}^k, \lambda^k)$ by an UCB $U_s^k(\boldsymbol{\lambda})$. Let $C_s^k = \sup_{\boldsymbol{\lambda} \in \Lambda}(U_s^k(\boldsymbol{\lambda}) - d(\hat{\mu}_{s-1}^k, \lambda^k))$.

$$\inf_{\boldsymbol{\lambda} \in \Lambda} \sum_{s=1}^t \sum_{k=1}^K w_s^k d(\hat{\mu}_{s-1}^k, \lambda^k) \geq \inf_{\boldsymbol{\lambda} \in \Lambda} \sum_{s=1}^t \sum_{k=1}^K w_s^k U_s^k(\boldsymbol{\lambda}) - \sum_{s=1}^t \sum_{k=1}^K w_s^k C_s^k \ .$$

Consider the following "optimistic" zero-sum games, indexed by $t \in \mathbb{N}$: to actions $(k, \boldsymbol{\lambda}) \in [K] \times \Lambda$ corresponds a reward of $U_t^k(\boldsymbol{\lambda})$ for the $k$-player.

An iterative saddle point algorithm attains an $(R_t^{\boldsymbol{\lambda}}, R_t^k)$ equilibrium at time $t$ on that sequence if

$$\inf_{\boldsymbol{\lambda} \in \Lambda} \sum_{s=1}^t \sum_{k=1}^K w_s^k U_s^k(\boldsymbol{\lambda}) + R_t^{\boldsymbol{\lambda}} \geq \sum_{s=1}^t \sum_{k=1}^K w_s^k \mathbb{E}_{\boldsymbol{\lambda} \sim \boldsymbol{q}_s} U_s^k(\boldsymbol{\lambda}) \geq \max_{k \in [K]} \sum_{s=1}^t \mathbb{E}_{\boldsymbol{\lambda} \sim \boldsymbol{q}_s} U_s^k(\boldsymbol{\lambda}) - R_t^k \ .$$

The notations $R_t^{\boldsymbol{\lambda}}$ and $R_t^k$ reflect a common strategy to attain such an equilibrium: instantiate two regret minimization algorithms for the two players, with linear losses $\ell_t^{\boldsymbol{w}}(\boldsymbol{w}) = -\mathbb{E}_{\boldsymbol{\lambda} \sim \boldsymbol{q}_t} \sum_{k=1}^K w^k U_t^k(\boldsymbol{\lambda})$ and $\ell_t^{\boldsymbol{\lambda}}(\boldsymbol{q}) = \mathbb{E}_{\boldsymbol{\lambda} \sim \boldsymbol{q}} \sum_{k=1}^K w_t^k U_t^k(\boldsymbol{\lambda})$. If we do so, the left and right inequalities are the regret properties of the algorithm for $\boldsymbol{\lambda}$ and $k$ respectively. At that point we have

$$\inf_{\boldsymbol{\lambda} \in \Lambda} \sum_{s=1}^t \sum_{k=1}^K w_s^k d(\hat{\mu}_{s-1}^k, \lambda^k) \geq \max_{k \in [K]} \sum_{s=1}^t \mathbb{E}_{\boldsymbol{\lambda} \sim \boldsymbol{q}_s} U_s^k(\boldsymbol{\lambda}) - R_t^{\boldsymbol{\lambda}} - R_t^k - \sum_{s=1}^t \sum_{k=1}^K w_s^k C_s^k \ .$$

We obtain the desired property with $x_t = R_t^{\boldsymbol{\lambda}} + R_t^k + \sum_{s=1}^t \sum_{k=1}^K w_s^k C_s^k$ .

**Use a regret property for $\boldsymbol{\lambda}$, then introduce UCBs, then use a regret property for $k$.** We take here for the $\boldsymbol{\lambda}$-player a regret minimization algorithm for the loss $\ell_t^{\boldsymbol{\lambda}}(\boldsymbol{q}) = \sum_{k=1}^K w_t^k \mathbb{E}_{\boldsymbol{\lambda} \sim \boldsymbol{q}} d(\hat{\mu}_{t-1}^k, \lambda^k)$. It verifies

$$\inf_{\boldsymbol{\lambda} \in \Lambda} \sum_{s=1}^t \sum_{k=1}^K w_s^k d(\hat{\mu}_{s-1}^k, \lambda^k) \geq \sum_{s=1}^t \sum_{k=1}^K w_s^k \mathbb{E}_{\boldsymbol{\lambda} \sim \boldsymbol{q}_s} d(\hat{\mu}_{s-1}^k, \lambda^k) - R_t^{\boldsymbol{\lambda}} \ .$$

We now introduce UCBs $U_s^k$,

$$\sum_{s=1}^{t}\sum_{k=1}^{K} w_s^k \, \mathbb{E}_{\boldsymbol{\lambda}\sim\boldsymbol{q}_s} \, d(\hat{\mu}_{s-1}^k, \lambda^k) \geq \sum_{s=1}^{t}\sum_{k=1}^{K} w_s^k U_s^k - \sum_{s=1}^{t}\sum_{k=1}^{K} w_s^k C_s^k \; .$$

The $k$-player uses a regret minimization algorithm for the loss $\ell_t^k(\boldsymbol{w}) = -\sum_{k=1}^{K} w_s^k U_s^k$, with regret $R_t^k$. Let $C_s^k = U_s^k - \mathbb{E}_{\boldsymbol{\lambda}\sim\boldsymbol{q}_s} \, d(\hat{\mu}_{t-1}^k, \lambda^k)$ .

$$\sum_{s=1}^{t}\sum_{k=1}^{K} w_s^k U_s^k \geq \max_k \sum_{s=1}^{t} U_s^k - R_t^k \; .$$

We obtain the desired property with $x_t = R_t^{\boldsymbol{\lambda}} + R_t^k + \sum_{s=1}^{t}\sum_{k=1}^{K} w_s^k C_s^k$ .

### D.3 Concentration arguments

Concentration event: $\mathcal{E}_t = \left\{ \forall s \leq t, \forall k \in [K], \; d(\hat{\mu}_s^k, \mu^k) \leq \frac{f(t^{1/(1+b)})}{N_s^k} \right\}$ .

**Lemma 14.** *Under the event $\mathcal{E}_t$, for all $s \in [t]$, $k \in [K]$ and $\boldsymbol{\lambda} \in \mathcal{M}$,*

$$|d(\mu^k, \lambda^k) - d(\hat{\mu}_{s-1}^k, \lambda^k)| \leq L\sqrt{2\sigma^2 \frac{f(t^{1/(1+b)})}{N_{s-1}^k}} \; .$$

*Proof.* Use the Lipschitz property of $x \mapsto d(x, y)$, then the sub-Gaussian assumption and finally the definition of $\mathcal{E}_t$. $\qquad\square$

**Lemma 15.** *Let $C_s^k = \max\left\{ 2L\sqrt{2\sigma^2 \frac{f(\max\{s-1, t^{1/(1+b)}\})}{N_{s-1}^k}}, \frac{f(\max\{s-1, t^{1/(1+b)}\})}{N_{s-1}^k} \right\}$. Let $\alpha_s^k$ and $\beta_s^k$ be defined as in section D.1. Under the event $\mathcal{E}_t$, for all $s \in [t]$, all $\boldsymbol{\lambda} \in \mathcal{M}$,*

$$\sup_{\xi\in[\alpha_s^k,\beta_s^k]} (U_s^k(\boldsymbol{\lambda}) - d(\xi, \lambda^k)) \leq C_s^k \; ,$$

$$\sup_{\xi\in[\alpha_s^k,\beta_s^k]} (U_s^k - \mathbb{E}_{\boldsymbol{\lambda}\sim\boldsymbol{q}_s} \, d(\xi, \lambda^k)) \leq C_s^k \; .$$

*Proof.* Let $u_s^k = \hat{\mu}_{s-1}^k - \sqrt{2\sigma^2 \frac{f(\max\{s-1, t^{1/(1+b)}\})}{N_{s-1}^k}}$ and $v_s^k = \hat{\mu}_{s-1}^k + \sqrt{2\sigma^2 \frac{f(\max\{s-1, t^{1/(1+b)}\})}{N_{s-1}^k}}$.

Under the event $\mathcal{E}_t$, for all $s \in [t]$, we have $\hat{\mu}_{s-1}^k, \mu^k \in [u_s^k, v_s^k]$. $U_s^k$ is defined as the maximum of $\frac{f(s-1)}{N_{s-1}^k}$ and a maximum over an interval which is contained in $[u_s^k, v_s^k]$. If $U_s^k$ is equal to the latter,

$$\sup_{\xi\in[\alpha_s^k,\beta_s^k]} (U_s^k - \mathbb{E}_{\boldsymbol{\lambda}\sim\boldsymbol{q}_s} \, d(\xi, \lambda^k)) \leq \sup_{\eta,\xi\in[u_s^k,v_s^k]} |\mathbb{E}_{\boldsymbol{\lambda}\sim\boldsymbol{q}_s} \, d(\eta, \lambda^k) - \mathbb{E}_{\boldsymbol{\lambda}\sim\boldsymbol{q}_s} \, d(\xi, \lambda^k)|$$

$$\leq L|u_s^k - v_s^k| \leq 2L\sqrt{2\sigma^2 \frac{f(\max\{s-1, t^{1/(1+b)}\})}{N_{s-1}^k}} \; .$$

If $U_s^k = \frac{f(s-1)}{N_{s-1}^k}$, then $\sup_{\xi\in[\alpha_s^k,\beta_s^k]}(U_s^k - \mathbb{E}_{\boldsymbol{\lambda}\sim\boldsymbol{q}_s} \, d(\xi, \lambda^k)) \leq U_s^k = \frac{f(s-1)}{N_{s-1}^k}$ .

Same computations for $U_s^k(\boldsymbol{\lambda})$, without expectations. $\qquad\square$

**Lemma 16.**

$$\sum_{s=K+1}^{t}\sum_{k=1}^{K} w_s^k C_s^k \leq 2L\sqrt{2\sigma^2 f(t)}(K^2 + 2\sqrt{2Kt}) + f(t)(K^2 + 2K\log(t/K)) \; .$$

*Proof.* Since $C_s^{\prime k}$ is the maximum of two quantities, it is smaller than their sum. By Lemma 9,

$$\sum_{s=K+1}^{t} \sum_{k=1}^{K} w_s^k 2L \sqrt{2\sigma^2 \frac{f(\max\{s-1, t^{1/(1+b)}\})}{N_{s-1}^k}} \leq 2L\sqrt{2\sigma^2 f(t)} \sum_{s=K+1}^{t} \sum_{k=1}^{K} \frac{w_s^k}{\sqrt{N_{s-1}^k}}$$

$$\leq 2L\sqrt{2\sigma^2 f(t)}(K^2 + 2\sqrt{2Kt}),$$

Similarly,

$$\sum_{s=K+1}^{t} \sum_{k=1}^{K} w_s^k \frac{f(\max\{s-1, t^{1/(1+b)}\})}{N_{s-1}^k} \leq f(t) \sum_{s=K+1}^{t} \sum_{k=1}^{K} \frac{w_s^k}{N_{s-1}^k}$$

$$\leq f(t)(K^2 + 2K \log(t/K)),$$

$\square$

**Lemma 17.** *Under $\mathcal{E}_t$, for any $\boldsymbol{\lambda} \in \mathcal{M}$,*

$$\sum_{k=1}^{K} N_t^k d(\hat{\mu}_t^k, \lambda^k) \geq \sum_{k=1}^{K} N_t^k d(\mu^k, \lambda^k) - L\sqrt{2\sigma^2 Kt f(t)}.$$

*Proof.* By the Lipschitzness assumption,

$$\sum_{k=1}^{K} N_t^k d(\hat{\mu}_t^k, \lambda^k) \geq \sum_{k=1}^{K} N_t^k d(\mu^k, \lambda^k) - L \sum_{k=1}^{K} N_t^k |\hat{\mu}_t^k - \mu^k|.$$

Using the sub-Gaussian hypothesis, under $\mathcal{E}_t$, $|\hat{\mu}_t^k - \mu^k| \leq \sqrt{2\sigma^2 d(\hat{\mu}_t^k, \mu)} \leq \sqrt{2\sigma^2 \frac{f(t)}{N_t^k}}.$

$$\sum_{k=1}^{K} N_t^k d(\hat{\mu}_t^k, \lambda^k) \geq \sum_{k=1}^{K} N_t^k d(\mu^k, \lambda^k) - L \sum_{k=1}^{K} N_t^k \sqrt{2\sigma^2 \frac{f(t)}{N_t^k}}$$

$$= \sum_{k=1}^{K} N_t^k d(\mu^k, \lambda^k) - L\sqrt{2\sigma^2 f(t)} \sum_{k=1}^{K} \sqrt{N_t^k}$$

$$\geq \sum_{k=1}^{K} N_t^k d(\mu^k, \lambda^k) - L\sqrt{2\sigma^2 Kt f(t)}.$$

$\square$

### D.4 The candidate answer

The data seen before time $t$ is summarized in the vector $\hat{\boldsymbol{\mu}}_{t-1} \in \Theta^K$. That vector does not in general belong to $\mathcal{M}$.

Our algorithm finds any point in the intersection of $\mathcal{M}$ and the confidence box around $\hat{\boldsymbol{\mu}}_{t-1}$. The point obtained is denoted by $\boldsymbol{\mu}_{t-1}^{\mathcal{M}}$ and verifies that for all $k \in [K]$, $d(\hat{\mu}_{t-1}^k, \mu_{t-1}^{\mathcal{M}k}) \leq \frac{f(t-1)}{N_{t-1}^k}$. The candidate answer used at time $t$ is then $i_t = i^*(\boldsymbol{\mu}_{t-1}^{\mathcal{M}})$.

### D.5 When the candidate answer is not the correct answer

**Chernoff information.** For $x, y \in \Theta$, let $\mathrm{ch}(x, y) = \inf_{u \in \Theta}(d(u, x) + d(u, y))$ be the Chernoff information between $x$ and $y$.

**Assumption 3.** There exists $\varepsilon > 0$ such that for all $\boldsymbol{\lambda} \in \neg i^*(\boldsymbol{\mu})$, there exists $k \in [K]$ such that $\mathrm{ch}(\lambda^k, \mu^k) \geq \varepsilon$.

If the distributions are sub-Gaussian with parameter $\sigma^2$, then $\mathrm{ch}(x, y) \geq \frac{(x-y)^2}{8\sigma^2}$ and that assumption is true for every $\boldsymbol{\mu} \in \mathcal{M}$ with $D_{\boldsymbol{\mu}} > 0$. i.e. Assumption 1 implies Assumption 3.

**Lemma 18.** *Suppose that Assumption 3 holds for $\boldsymbol{\mu} \in \mathcal{M}$ and that for all $k \in [K]$, $d(\hat{\mu}^k_{t-1}, \mu^k) \le \frac{\log(t-1)}{N^k_{t-1}}$. If $i^*(\boldsymbol{\mu}^{\mathcal{M}}_{t-1}) \ne i^*(\boldsymbol{\mu})$ then there exists $j \in [K]$ such that $\frac{f(t-1)}{N^j_{t-1}} \ge \frac{\varepsilon}{2}$.*

*Proof.* If $i^*(\tilde{\boldsymbol{\mu}}_{t-1}) \ne i^*(\boldsymbol{\mu})$ then $\boldsymbol{\mu}^{\mathcal{M}}_{t-1}$ belongs to the set $\neg i^*(\boldsymbol{\mu})$.

By Assumption 3, there exists $j \in [K]$ such that $\mathrm{ch}(\mu^k, \mu^{\mathcal{M}}_{t-1}{}^k) \ge \varepsilon$. By definition of $\mathrm{ch}$ as an infimum over $\Theta$, it is smaller than $d(\hat{\mu}^j_{t-1}, \mu^j) + d(\hat{\mu}^j_{t-1}, \mu^{\mathcal{M}}_{t-1}{}^j)$. That sum is then bigger than $\varepsilon$, with consequence that either $d(\hat{\mu}^j_{t-1}, \mu^j) \ge \varepsilon/2$ or $d(\hat{\mu}^j_{t-1}, \mu^{\mathcal{M}}_{t-1}{}^j) \ge \varepsilon/2$.

If $d(\hat{\mu}^j_{t-1}, \mu^j) \ge \varepsilon/2$, then by hypothesis, $\frac{f(t-1)}{N^j_{t-1}} \ge d(\hat{\mu}^j_{t-1}, \mu^j) \ge \varepsilon/2$.

Otherwise $d(\hat{\mu}^j_{s-1}, \mu^{\mathcal{M}}_{t-1}{}^j) \ge \varepsilon/2$. By definition of $\mu^{\mathcal{M}}_{t-1}$, $\frac{f(t-1)}{N^j_{t-1}} \ge d(\hat{\mu}^j_{t-1}, \mu^{\mathcal{M}}_{t-1}{}^j)$. We proved that $\frac{f(t-1)}{N^j_{t-1}} \ge \frac{\varepsilon}{2}$. $\qquad\square$

**Linear increase in information.** For $i \in \mathcal{I}$, let $n_i(t)$ be the number of stages $s \le t$ in which $i_s = i$. To shorten notations, let $i^* = i^*(\boldsymbol{\mu})$. The goal of this section is to find a lower bound for $n_{i^*}(t)$. We do it by showing that when the answer $i_s$ is not the correct one, a quantity is linearly increasing, while at the same time being $O(\sqrt{t})$ by a concentration argument. Hence the number of time steps this can happen is also $O(\sqrt{t})$.

Using that $\boldsymbol{\mu} \in \neg i_s$,

$$\sum_{s \le t, i_s \ne i^*} \sum_{k=1}^{K} w^k_s d(\hat{\mu}^k_{s-1}, \mu^k) \ge \sum_{i \in \mathcal{I} \setminus \{i^*\}} \inf_{\boldsymbol{\lambda} \in \neg i} \sum_{s \le t, i_s = i} \sum_{k=1}^{K} w^k_s d(\hat{\mu}^k_{s-1}, \lambda^k) \, .$$

Let $\varepsilon_t$ be the quantity on the left, which will be small by a concentration argument.

The algorithm used when $i_s = i$ is an optimistic approximate saddle point algorithm with slack $R^k_{n_i(t)} + R^{\boldsymbol{\lambda}}_{n_i(t)} + \sum_{s \le t, i_s = i} \sum_{k=1}^{K} w^k_s C^k_s$. Hence we have

$$\varepsilon_t \ge \sum_{i \in \mathcal{I} \setminus \{i^*\}} \max_k \sum_{s \le t, i_s = i} U^k_s - \sum_{i \in \mathcal{I} \setminus \{i^*\}} (R^k_{n_i(t)} + R^{\boldsymbol{\lambda}}_{n_i(t)}) - \sum_{s \le t, i_s \ne i^*} \sum_{k=1}^{K} w^k_s C^k_s \, .$$

Note: if UCBs of the form $U^k_s(\boldsymbol{\lambda})$ are used instead of $U^k_s$, replace $U^k_s$ by $\mathbb{E}_{\boldsymbol{\lambda} \sim \boldsymbol{q}_s} U^k_s(\boldsymbol{\lambda})$ here and in the following expressions.

For fixed $i \in \mathcal{I} \setminus \{i^*\}$, we now shos that the quantity $\max_k \sum_{s \le t, i_s = i} U^k_s$ increases linearly with the number of terms of the sum, $n_i(t)$. We proved in Lemma 13 that for all $s \in \mathbb{N}$ and $k \in [K]$, $U^k_s \ge \frac{f(s-1)}{N^k_{s-1}}$. When the event $\mathcal{E}_t$ holds, for all $s \in [t^{1/(1+b)}, t]$ with $i_s \ne i^*$, there is a $j_s \in [K]$ such that $U^{j_s}_s \ge \varepsilon/2$ by Lemma 18.

Let $t'$ be the last term of the sum and suppose that $t' > \sqrt{t}$. Let $j$ be such that $U^j_{t'} \ge \varepsilon/2$. Then for all $s \in [\lceil t^{1/(1+b)} \rceil, t']$,

$$\frac{f(s-1)}{N^j_{s-1}} \ge \frac{f(s-1)}{N^j_{t'-1}} = \frac{f(s-1)}{f(t'-1)} \frac{f(t'-1)}{N^j_{t'-1}} \ge \frac{f(t^{1/(1+b)})}{f(t)} \varepsilon/2 \, .$$

For $t > e$, $\frac{f(t^{1/(1+b)})}{f(t)} \ge \frac{1}{3(1+b)}$. Let $C_b = 1/(3(1+b))$.

Hence for that arm $j$, $\sum_{s \le t, i_s = i} U^j_s \ge C_b \varepsilon (n_i(t) - n_i(t^{1/(1+b)}))/2$.

We conclude that the maximum over $k$ of the sums is also bigger than this quantity. We have shown

$$\varepsilon_t \geq \sum_{i\in\mathcal{I}\setminus\{i^*\}} \frac{C_b\varepsilon}{2}(n_i(t) - n_i(t^{1/(1+b)})) - \sum_{i\in\mathcal{I}\setminus\{i^*\}}(R^k_{n_i(t)} + R^{\boldsymbol{\lambda}}_{n_i(t)}) - \sum_{s=K+1}^{t}\sum_{k=1}^{K} w^k_s C^k_s$$

$$\geq \frac{C_b\varepsilon}{2}(t - t^{1/(1+b)} - n_{i^*}(t)) - (|\mathcal{I}| - 1)(R^k_t + R^{\boldsymbol{\lambda}}_t) - \sum_{s=K+1}^{t}\sum_{k=1}^{K} w^k_s C^k_s\,.$$

If $n \mapsto R^k_n$ and $n \mapsto R^{\boldsymbol{\lambda}}_n$ are concave (for example regret proportional to $\sqrt{n}$), the regret term has the form $(|\mathcal{I}| - 1)(R^k_{(t-n_{i^*})/(|I|-1)} + R^{\boldsymbol{\lambda}}_{(t-n_{i^*})/(|I|-1)})$.

By concentration,

$$\varepsilon_t \leq f(t^{1/(1+b)}) \sum_{s=1}^{t}\sum_{k=1}^{K} \frac{w^k_s}{N^t_{s-1}} \leq f(t)(K^2 + 2K\log(t/K))\,.$$

We proved

$$n_{i^*}(t) \geq t - t^{1/(1+b)} - \frac{2}{C_b\varepsilon}\left((|\mathcal{I}| - 1)(R^k_t + R^{\boldsymbol{\lambda}}_t) + f(t)(K^2 + 2K\log(t/K)) + \sum_{s=K+1}^{t}\sum_{k=1}^{K} w^k_s C^k_s\right)$$

$$(2)$$

### D.6   When the candidate answer is the correct answer

Let $t' \leq t$ be the last round in which $i_{t'} = i^*$ before the algorithm stops. Then $t' \geq n_{i^*}(t)$, we have $i_{t'} = i^*$ and $n_{i^*}(t') = n_{i^*}(t)$.

$$\beta(t,\delta) \geq \beta(t',\delta) \geq \inf_{\boldsymbol{\lambda}\in\neg i_{t'}} \sum_{k=1}^{K} N^k_{t'} d(\hat{\mu}^k_{t'}, \lambda^k) = \inf_{\boldsymbol{\lambda}\in\neg i^*} \sum_{k=1}^{K} N^k_{t'} d(\hat{\mu}^k_{t'}, \lambda^k)$$

$$\geq \inf_{\boldsymbol{\lambda}\in\neg i^*} \sum_{k=1}^{K} N^k_{t'} d(\mu^k, \lambda^k) - L\sqrt{2\sigma^2 K t f(t)}\,.$$

Using the tracking Lemma 7, then concentration Lemma 14,

$$\beta(t,\delta) \geq \inf_{\boldsymbol{\lambda}\in\neg i^*} \sum_{s=1}^{t'}\sum_{k=1}^{K} w^k_s d(\mu^k, \lambda^k) - KD - L\sqrt{2\sigma^2 K t f(t)}$$

$$\geq \inf_{\boldsymbol{\lambda}\in\neg i^*} \sum_{s=K+1}^{t'}\sum_{k=1}^{K} w^k_s d(\hat{\mu}^k_{s-1}, \lambda^k)$$

$$- L\sqrt{2\sigma^2 f(t)} \sum_{s=K+1}^{t}\sum_{k=1}^{K} \frac{w^k_s}{\sqrt{N^k_{s-1}}} - KD - L\sqrt{2\sigma^2 K t f(t)}$$

$$\geq \inf_{\boldsymbol{\lambda}\in\neg i^*} \sum_{s=K+1}^{t'}\sum_{k=1}^{K} w^k_s d(\hat{\mu}^k_{s-1}, \lambda^k)$$

$$- 2L\sqrt{2\sigma^2 f(t)}(K^2 + 2\sqrt{2Kt}) - KD - L\sqrt{2\sigma^2 K t f(t)}$$

We drop the rounds in which $i_s \neq i^*$.

$$\beta(t,\delta) \geq \inf_{\boldsymbol{\lambda}\in\neg i^*} \sum_{K+1\leq s\leq t', i_s = i^*}\sum_{k=1}^{K} w^k_s d(\hat{\mu}^k_{s-1}, \lambda^k)$$

$$- 2L\sqrt{2\sigma^2 f(t)}(K^2 + 2\sqrt{2Kt}) - KD - L\sqrt{2\sigma^2 K t f(t)}\,.$$

The algorithm used is an optimistic approximate saddle point algorithm with slack $R_t^{\boldsymbol{\lambda}} + R_t^k + \sum_{s=1}^{t} \sum_{k=K+1}^{K} w_s^k C_s^k$ :

$$\inf_{\boldsymbol{\lambda} \in \neg i^*} \sum_{s \leq t', i_s = i^*} \sum_{k=1}^{K} w_s^k d(\hat{\mu}_{s-1}^k, \lambda^k) \geq \max_k \sum_{s \leq t', i_s = i^*} U_s^k - \left(R_t^{\boldsymbol{\lambda}} + R_t^k + \sum_{s=K+1}^{t} \sum_{k=1}^{K} w_s^k C_s^k\right) .$$

Let $A_t = \sum_{s=K+1}^{t} \sum_{k=1}^{K} w_s^k C_s^k + 2L\sqrt{2\sigma^2 f(t)}(K^2 + 2\sqrt{2Kt}) + KD + L\sqrt{2\sigma^2 Kt f(t)}$. We obtain

$$\beta(t,\delta) \geq \max_k \sum_{K < s \leq t', i_s = i^*} U_s^k - R_t^k - R_t^{\boldsymbol{\lambda}} - A_t .$$

Let $t_b = t^{1/(1+b)}$. Since $U_t$ is a coordinate-wise upper confidence bound when concentration holds (for $s \geq t^{1/(1+b)}$), we have

$$\beta(t,\delta) \geq \max_k \sum_{t_b \leq s \leq t', i_s = i^*} \mathbb{E}_{\boldsymbol{\lambda} \sim \boldsymbol{q}_s} d(\mu^k, \lambda^k) - R_t^k - R_t^{\boldsymbol{\lambda}} - A_t$$

$$= (n_{i^*}(t') - t_b) \max_k \frac{1}{(n_{i^*}(t') - t_b)} \sum_{t^{1/(1+b)} \leq s \leq t', i_s = i^*} \mathbb{E}_{\boldsymbol{\lambda} \sim \boldsymbol{q}_s} d(\mu^k, \lambda^k) - R_t^k - R_t^{\boldsymbol{\lambda}} - A_t$$

$$\geq (n_{i^*}(t') - t_b) \inf_{\boldsymbol{q} \in \mathbb{P}(\neg i^*)} \max_k \mathbb{E}_{\boldsymbol{\lambda} \sim \boldsymbol{q}} d(\mu^k, \lambda^k) - R_t^k - R_t^{\boldsymbol{\lambda}} - A_t$$

$$= (n_{i^*}(t') - t^{1/(1+b)}) D_{\boldsymbol{\mu}} - R_t^k - R_t^{\boldsymbol{\lambda}} - A_t .$$

$t'$ is such that $n_{i^*}(t') = n_{i^*}(t)$. Combining that result and the lower bound on $n_{i^*}(t)$ of equation (2), we have

$$\frac{\beta(t,\delta) + A_t + R_t^k + R_t^{\boldsymbol{\lambda}}}{D_{\boldsymbol{\mu}}} \tag{3}$$

$$\geq t - 2t^{1/(1+b)} - \frac{2}{C_b \varepsilon} \left( (|\mathcal{I}| - 1)(R_t^k + R_t^{\boldsymbol{\lambda}}) + f(t)(K^2 + 2K\log(t/K)) + \sum_{s=K+1}^{t} \sum_{k=1}^{K} w_s^k C_s^k \right) . \tag{4}$$

### D.7 Stopping time upper bound

We can solve equation (3) to find an upper bound for $t$ such that the algorithm does not stop. Suppose that there exists $R > 0$ such that $R_t^k + R_t^{\boldsymbol{\lambda}} \leq R\sqrt{Kt}$. Take $b = 1$. By Lemma 16,

$\sum_{s=K+1}^{t} \sum_{k=1}^{K} w_s^k C_s^k \leq 2L\sqrt{2\sigma^2 f(t)}(K^2 + 2\sqrt{2Kt}) + f(t)(K^2 + 2K\log(t/K))$.

$A_t \leq 4L\sqrt{2\sigma^2 f(t)}(K^2 + 2\sqrt{2Kt}) + KD + L\sqrt{2\sigma^2 Kt f(t)} + f(t)(K^2 + 2K\log(t/K))$.

We now define

$$h(t) = 2\sqrt{t} + \frac{A_t + R\sqrt{Kt}}{D_{\boldsymbol{\mu}}}$$

$$+ \frac{2}{C_b \varepsilon} \left( (|\mathcal{I}| - 1)R\sqrt{Kt} + 2f(t)(K^2 + 2K\log(t/K)) + 2L\sqrt{2\sigma^2 f(t)}(K^2 + 2\sqrt{2Kt}) \right) .$$

We have that $h(t) = \mathcal{O}(\sqrt{t \log t})$ and we obtained that if $t < \tau_\delta$ then

$$t - h(t) \leq \frac{\beta(t,\delta)}{D_{\boldsymbol{\mu}}} .$$

## E Algorithms

### E.1 Optimistic Track and Stop

We prove that under the concentration event $\mathcal{E}_t$, there is an upper bound on $t$ such that $t < \tau_\delta$.

Let $\mathcal{C}_s = \{\boldsymbol{\xi} \in \Theta^K : \forall k \in [K], d(\hat{\mu}_{s-1}^k, \xi^k) \leq \frac{f(s-1)}{N_{s-1}^k}\}$ be a confidence region around $\hat{\boldsymbol{\mu}}_{s-1}$.

**When $i_t \neq i^*(\boldsymbol{\mu})$.** Let $i \in \mathcal{I} \setminus \{i^*(\boldsymbol{\mu})\}$. Since $i_s \neg i^*(\boldsymbol{\mu})$ implies that $\boldsymbol{\mu} \in \neg i_s$,

$$\sum_{s \leq t, i_s = i} \sum_{k=1}^{K} w_s^k d(\hat{\mu}_{s-1}^k, \mu^k) \geq \inf_{\boldsymbol{\lambda} \in \neg i} \sum_{s \leq t, i_s = i} \sum_{k=1}^{K} w_s^k d(\hat{\mu}_{s-1}^k, \lambda^k) \, .$$

Let $\varepsilon_t^i$ be the left hand side of that inequality. Since $\hat{\boldsymbol{\mu}}_{s-1}$ and $\boldsymbol{\mu}_s^+$ both belong to $\mathcal{C}_s$, we have

$$\sum_{s \leq t, i_s = i} \sum_{k=1}^{K} w_s^k d(\hat{\mu}_{s-1}^k, \lambda^k) \geq \sum_{s \leq t, i_s = i} \sum_{k=1}^{K} w_s^k d(\mu_s^{+k}, \lambda^k) - L\sqrt{2\sigma^2 f(t)} \sum_{s \leq t, i_s = i} \frac{w_s^k}{\sqrt{N_{s-1}^k}} \, .$$

By definition of $\boldsymbol{\mu}_s^+$,

$$\inf_{\boldsymbol{\lambda} \in \neg i} \sum_{s \leq t, i_s = i} \sum_{k=1}^{K} w_s^k d(\mu_s^{+k}, \lambda^k) \geq \sum_{s \leq t, i_s = i} \inf_{\boldsymbol{\lambda} \in \neg i} \sum_{k=1}^{K} w_s^k d(\mu_s^{+k}, \lambda^k) = \sum_{s \leq t, i_s = i} D_{\boldsymbol{\mu}_s^+} \, .$$

For $s \geq t^{1/(1+b)}$, $\boldsymbol{\mu} \in \mathcal{C}_s$ and by definition of $\boldsymbol{\mu}_s^+$, $D_{\boldsymbol{\mu}_s^+} \geq D_{\boldsymbol{\mu}}$. We obtain, with $n_i(t)$ the number of times with $i_s = i$ until $t$,

$$\sum_{i \in \mathcal{I} \setminus \{i^*(\boldsymbol{\mu})\}} \varepsilon_t^i \geq (t - n_{i^*(\boldsymbol{\mu})}(t) - t^{1/(1+b)}) D_{\boldsymbol{\mu}} - L\sqrt{2\sigma^2 f(t)} \sum_{s \leq t} \frac{w_s^k}{\sqrt{N_{s-1}^k}}$$

$$\geq (t - n_{i^*(\boldsymbol{\mu})}(t) - t^{1/(1+b)}) D_{\boldsymbol{\mu}} - L\sqrt{2\sigma^2 f(t)}(K^2 + 2\sqrt{2Kt}) \, .$$

See Lemma 9 for that last inequality. By concentration,

$$\sum_{i \in \mathcal{I} \setminus \{i^*(\boldsymbol{\mu})\}} \varepsilon_t^i \leq f(t) \sum_{s=1}^{t} \sum_{k=1}^{K} \frac{w_s^k}{N_{s-1}^k} \leq f(t)(K^2 + 2K \log(t/K)) \, .$$

Finally,

$$n_{i^*(\boldsymbol{\mu})}(t) \geq t - t^{1/(1+b)} - \frac{1}{D_{\boldsymbol{\mu}}} \left( L\sqrt{2\sigma^2 f(t)}(K^2 + 2\sqrt{2Kt}) + f(t)(K^2 + 2K \log(t/K)) \right) \, .$$

**When $i_t = i^*(\boldsymbol{\mu})$.** Let $t' \geq n_{i^*(\boldsymbol{\mu})}(t)$ be such that $i_{t'} = i^*(\boldsymbol{\mu})$ and $n_{i^*(\boldsymbol{\mu})}(t') = n_{i^*(\boldsymbol{\mu})}(t)$. Using concentration and tracking properties, as in the main sample complexity proof of Appendix D.6,

$$\beta(t', \delta) \geq \inf_{\boldsymbol{\lambda} \in \neg i^*(\boldsymbol{\mu})} \sum_{k=1}^{K} N_t^k d(\hat{\mu}_{t'}^k, \lambda^k)$$

$$\geq \inf_{\boldsymbol{\lambda} \in \neg i^*(\boldsymbol{\mu})} \sum_{k=1}^{K} N_t^k d(\mu^k, \lambda^k) - L\sqrt{2\sigma^2 Kt f(t)}$$

$$\geq \inf_{\boldsymbol{\lambda} \in \neg i^*(\boldsymbol{\mu})} \sum_{s=1}^{t'} \sum_{k=1}^{K} w_s^k d(\mu^k, \lambda^k) - KD - L\sqrt{2\sigma^2 Kt f(t)}$$

$$\geq \inf_{\boldsymbol{\lambda} \in \neg i^*(\boldsymbol{\mu})} \sum_{s=1}^{t'} \sum_{k=1}^{K} w_s^k d(\hat{\mu}_{s-1}^k, \lambda^k)$$

$$- L\sqrt{2\sigma^2 f(t)}(K^2 + 2\sqrt{2Kt}) - KD - L\sqrt{2\sigma^2 Kt f(t)}$$

Since $\hat{\boldsymbol{\mu}}_{s-1}$ and $\boldsymbol{\mu}_s^+$ both belong to $\mathcal{C}_s$, we have

$$\inf_{\boldsymbol{\lambda} \in \neg i^*(\boldsymbol{\mu})} \sum_{s=1}^{t'} \sum_{k=1}^{K} w_s^k d(\hat{\mu}_{s-1}^k, \lambda^k) \geq \inf_{\boldsymbol{\lambda} \in \neg i^*(\boldsymbol{\mu})} \sum_{s=1}^{t'} \sum_{k=1}^{K} w_s^k d(\mu_s^{+k}, \lambda^k) - L\sqrt{2\sigma^2 f(t)}(K^2 + 2\sqrt{2Kt}) \, .$$

Let $B_t = 2L\sqrt{2\sigma^2 f(t)}(K^2 + 2\sqrt{2Kt}) + KD + L\sqrt{2\sigma^2 Kt f(t)}$.

$$\beta(t,\delta) \geq \inf_{\boldsymbol{\lambda} \in \neg i^*(\boldsymbol{\mu})} \sum_{s \leq t', i_s = i^*(\boldsymbol{\mu})} \sum_{k=1}^{K} w_s^k d(\mu_s^{+k}, \lambda^k) - B_t$$

$$\geq \sum_{s \leq t', i_s = i^*(\boldsymbol{\mu})} \inf_{\boldsymbol{\lambda} \in \neg i_t} \sum_{k=1}^{K} w_s^k d(\mu_s^{+k}, \lambda^k) - B_t$$

$$= \sum_{s \leq t', i_s = i^*(\boldsymbol{\mu})} D_{\boldsymbol{\mu}_s^+} - B_t \ .$$

For $s \geq t^{1/(1+b)}$, $\boldsymbol{\mu} \in \mathcal{C}_s$. Then by definition of $\boldsymbol{\mu}_s^+$, $D_{\boldsymbol{\mu}_s^+} \geq D_{\boldsymbol{\mu}}$.

$$\beta(t,\delta) \geq \sum_{t^{1/(1+b)} \leq s \leq t', i_s = i^*(\boldsymbol{\mu})} D_{\boldsymbol{\mu}} - B_t$$

$$= (n_{i^*(\boldsymbol{\mu})}(t) - t^{1/(1+b)}) D_{\boldsymbol{\mu}} - B_t \ .$$

**Putting things together.** Let $h(t) = 3L\sqrt{2\sigma^2 f(t)}(K^2 + 2\sqrt{2Kt}) + f(t)(K^2 + 2K\log(t/K)) + KD + L\sqrt{2\sigma^2 Kt f(t)}$. When the concentration event $\mathcal{E}_t$ holds, if $t < \tau_\delta$ then

$$\frac{\beta(t,\delta) + h(t)}{D_{\boldsymbol{\mu}}} \geq t - 2t^{1/(1+b)} \ .$$

Let $T_0(\delta)$ be the maximal $t$ verifying this inequality. Then the expected sample complexity is lower than $T_0(\delta) + \frac{2eK}{a^2}$. Note that $f(t)$ depends on $a$ and $b$.

### E.2 Follow The Perturbed Leader

In this section, we suppose that the rewards are bounded and we define $C > 0$ such that for all times $s$ and $k \in [K]$, $|X_s^k - \hat{\mu}_{s-1}^k| \leq C$.

At stage $t$, the loss of a vector $\boldsymbol{\lambda}$ is $\ell_t(\boldsymbol{\lambda}) = d(\hat{\mu}_{t-1}^{k_t}, \lambda^{k_t})$. The only unknown quantity for the $\boldsymbol{\lambda}$-player is $k_t$. We will use the form of that loss in the way we perturb the leader. For $\boldsymbol{\sigma} \in \mathbb{R}_+^K$ and $\boldsymbol{\xi} \in \Theta^K$ we define

$$\boldsymbol{\lambda}_t(\boldsymbol{\sigma}, \boldsymbol{\xi}) = \operatorname*{argmin}_{\boldsymbol{\lambda}} \sum_{s=1}^{t-1} \ell_s(\boldsymbol{\lambda}) + \sum_{k=1}^{K} \sigma^k d(\xi^k, \lambda^k) \ .$$

We study the expected regret of an algorithm playing $\boldsymbol{\lambda}_t(\boldsymbol{\sigma}_t, \hat{\boldsymbol{\mu}}_{t-1})$ with exponentially distributed perturbations $\boldsymbol{\sigma}_t$. Let $\boldsymbol{q}_t$ be the distribution of $\boldsymbol{\lambda}_t(\boldsymbol{\sigma}_t, \hat{\boldsymbol{\mu}}_{t-1})$. Let $\tilde{\mu}_{t-1}^k = \frac{1}{N_{t-1}^k} \sum_{s=1}^{t-1} \hat{\mu}_{s-1}^k \mathbb{I}\{k_s = k\}$. We show in the following lemma that the point $\boldsymbol{\lambda}_t(\boldsymbol{\sigma}_t, \hat{\boldsymbol{\mu}}_{t-1})$ can be computed by the best-response oracle, as

$$\operatorname*{argmin}_{\boldsymbol{\lambda} \in \Lambda} \sum_{k=1}^{K} (N_{t-1}^k + \sigma_t^k) d\left(\frac{N_{t-1}^k}{N_{t-1}^k + \sigma_t^k} \tilde{\mu}_{t-1}^k + \frac{\sigma_t^k}{N_{t-1}^k + \sigma_t^k} \hat{\mu}_{t-1}^k, \lambda^k\right) \ .$$

**Lemma 19.** *Let $(\boldsymbol{\mu}_s)_{s \in [t]}$ be $t$ points in $\Theta^K$. Then*

$$\operatorname*{argmin}_{\boldsymbol{\lambda} \in \Lambda} \sum_{s=1}^{t} d(\mu_s^{k_s}, \lambda^{k_s}) = \operatorname*{argmin}_{\boldsymbol{\lambda} \in \Lambda} \sum_{k=1}^{K} N_t^k d\left(\frac{\sum_{s=1}^{t} \mu_s^k \mathbb{I}\{k_s = k\}}{N_t^k}, \lambda^k\right) \ .$$

*Proof.* This is an extension of the following property:

$$\operatorname*{argmin}_{\lambda} d(\mu_1, \lambda) + d(\mu_2, \lambda) = \operatorname*{argmin}_{\lambda} d\left(\frac{\mu_1 + \mu_2}{2}, \lambda\right) \ .$$

Indeed we can observe that fact by developing the divergence in terms of $\phi$ and observing that the terms depending on $\lambda$ are the same up to a multiplicative factor.

$$d(\mu_1, \lambda) + d(\mu_2, \lambda) = \phi(\mu_1) + \phi(\mu_2) - 2\left(\phi(\lambda) + \phi'(\lambda)(\frac{\mu_1 + \mu_2}{2} - \lambda)\right) ,$$

$$d(\frac{\mu_1 + \mu_2}{2}, \lambda) = \phi(\frac{\mu_1 + \mu_2}{2}) - \left(\phi(\lambda) + \phi'(\lambda)(\frac{\mu_1 + \mu_2}{2} - \lambda)\right) .$$

$\square$

**Theorem 3.** *The expected regret of the FTPL procedure introduced above against an oblivious adversary, with perturbations $\sigma_t^k = \eta_t^k \sigma_1^k$ with $\eta_t^k = \sqrt{N_{t-1}^k}$ and $\sigma_1^k$ exponential with parameter $\eta$ is*

$$\sum_{s=1}^{t} \mathbb{E}_{\boldsymbol{\lambda} \sim q_s} \ell_s(\boldsymbol{\lambda}) - \inf_{\boldsymbol{\lambda} \in \Lambda} \sum_{s=1}^{t} \ell_s(\boldsymbol{\lambda}) \le R_t = \sqrt{Kt}\left(\frac{D + 2CL}{\eta} + 2D\eta\right) .$$

The expected regret of the FTPL algorithm in which the noises are independent in time and $\sigma_t^k$ is exponential with parameter $\eta/\eta_t^k$ is the same.

For non-oblivious adversaries, the quantity $\sum_{s=1}^{t} \mathbb{E}_{\boldsymbol{\lambda} \sim q_s} \ell_s(\boldsymbol{\lambda}) - \inf_{\boldsymbol{\lambda} \in \Lambda} \sum_{s=1}^{t} \ell_s(\boldsymbol{\lambda})$ is also bounded by the same $R_t$, according to Lemma 4.1 of [4].

*Proof of Theorem 3.* Regret decomposition: for any $u$, the regret compared to $u$ is

$$\sum_{s=1}^{t} \ell_s(\boldsymbol{\lambda}_s(\boldsymbol{\sigma}_s, \hat{\boldsymbol{\mu}}_{s-1})) - \sum_{s=1}^{t} \ell_s(u) \le \sum_{s=1}^{t} \ell_s(\boldsymbol{\lambda}_s(\boldsymbol{\sigma}_s, \hat{\boldsymbol{\mu}}_{s-1})) - \ell_s(\boldsymbol{\lambda}_{s+1}(\boldsymbol{\sigma}_s, \hat{\boldsymbol{\mu}}_{s-1}))$$

$$+ \sum_{s=1}^{t} \ell_s(\boldsymbol{\lambda}_{s+1}(\boldsymbol{\sigma}_s, \hat{\boldsymbol{\mu}}_{s-1})) - \sum_{s=1}^{t} \ell_s(u)$$

**Second term of the regret.** We are analysing here the regret of a noisy Be-The-Leader. We first show by induction that

$$\sum_{s=1}^{t} \ell_s(\boldsymbol{\lambda}_{s+1}(\boldsymbol{\sigma}_s, \hat{\boldsymbol{\mu}}_{s-1})) - \sum_{s=1}^{t} \ell_s(u)$$

$$\le \boldsymbol{\sigma}_t^\mathsf{T} d(\hat{\boldsymbol{\mu}}_{t-1}, u) + \sum_{s=1}^{t} \boldsymbol{\sigma}_s^\mathsf{T} d(\hat{\boldsymbol{\mu}}_{s-1}, \boldsymbol{\lambda}_{s+1}(\boldsymbol{\sigma}_s, \hat{\boldsymbol{\mu}}_{s-1})) - \boldsymbol{\sigma}_{s-1}^\mathsf{T} d(\hat{\boldsymbol{\mu}}_{s-2}, \boldsymbol{\lambda}_{s+1}(\boldsymbol{\sigma}_s, \hat{\boldsymbol{\mu}}_{s-1}))$$

Initialization: for all $u \in \Lambda$,

$$\ell_1(\boldsymbol{\lambda}_2(\boldsymbol{\sigma}_1, \hat{\boldsymbol{\mu}}_0)) = \ell_1(\boldsymbol{\lambda}_2(\boldsymbol{\sigma}_1, \hat{\boldsymbol{\mu}}_0)) + \boldsymbol{\sigma}_1^\mathsf{T} d(\hat{\boldsymbol{\mu}}_0, \boldsymbol{\lambda}_2(\boldsymbol{\sigma}_1, \hat{\boldsymbol{\mu}}_0)) - \boldsymbol{\sigma}_1^\mathsf{T} d(\hat{\boldsymbol{\mu}}_0, \boldsymbol{\lambda}_2(\boldsymbol{\sigma}_1, \hat{\boldsymbol{\mu}}_0))$$

$$\le \ell_1(u) + \boldsymbol{\sigma}_1^\mathsf{T} d(\hat{\boldsymbol{\mu}}_0, u) - \boldsymbol{\sigma}_1^\mathsf{T} d(\hat{\boldsymbol{\mu}}_0, \boldsymbol{\lambda}_2(\boldsymbol{\sigma}_1, \hat{\boldsymbol{\mu}}_0)) .$$

Let $A_1(u) = \boldsymbol{\sigma}_1^\mathsf{T} d(\hat{\boldsymbol{\mu}}_0, u) - \boldsymbol{\sigma}_1^\mathsf{T} d(\hat{\boldsymbol{\mu}}_0, \boldsymbol{\lambda}_2(\boldsymbol{\sigma}_1, \hat{\boldsymbol{\mu}}_0))$. Then for all $u \in \Lambda$, $\ell_1(\boldsymbol{\lambda}_2(\boldsymbol{\sigma}_1, \hat{\boldsymbol{\mu}}_0)) - \ell_1(u) \le A_1(u)$.

Induction: suppose that for all $u \in \Lambda$, $\sum_{s=1}^{t-1} \ell_s(\boldsymbol{\lambda}_{s+1}(\boldsymbol{\sigma}_s, \hat{\boldsymbol{\mu}}_{s-1})) \le \sum_{s=1}^{t-1} \ell_s(u) + A_{t-1}(u)$, with

$$A_{t-1}(u) = \boldsymbol{\sigma}_{t-1}^\mathsf{T} d(\hat{\boldsymbol{\mu}}_{t-2}, u) + \sum_{s=1}^{t-1} \boldsymbol{\sigma}_{s-1}^\mathsf{T} d(\hat{\boldsymbol{\mu}}_{s-2}, \boldsymbol{\lambda}_{s+1}(\boldsymbol{\sigma}_s, \hat{\boldsymbol{\mu}}_{s-1})) - \boldsymbol{\sigma}_s^\mathsf{T} d(\hat{\boldsymbol{\mu}}_{s-1}, \boldsymbol{\lambda}_{s+1}(\boldsymbol{\sigma}_s, \hat{\boldsymbol{\mu}}_{s-1}))$$

where $\boldsymbol{\sigma}_0 = 0$. Apply it to $u = \boldsymbol{\lambda}_{t+1}(\boldsymbol{\sigma}_t, \hat{\boldsymbol{\mu}}_{t-1})$.

$$
\begin{aligned}
\sum_{s=1}^{t} \ell_s(\boldsymbol{\lambda}_{s+1}(\boldsymbol{\sigma}_s, \hat{\boldsymbol{\mu}}_{s-1})) &\leq \sum_{s=1}^{t-1} \ell_s(\boldsymbol{\lambda}_{t+1}(\boldsymbol{\sigma}_t, \hat{\boldsymbol{\mu}}_{t-1})) + \ell_t(\boldsymbol{\lambda}_{t+1}(\boldsymbol{\sigma}_t, \hat{\boldsymbol{\mu}}_{t-1})) \\
&\quad + A_{t-1}(\boldsymbol{\lambda}_{t+1}(\boldsymbol{\sigma}_t, \hat{\boldsymbol{\mu}}_{t-1})) \\
&= \sum_{s=1}^{t} \ell_s(\boldsymbol{\lambda}_{t+1}(\boldsymbol{\sigma}_t, \hat{\boldsymbol{\mu}}_{t-1})) + \boldsymbol{\sigma}_t^{\mathsf{T}} d(\hat{\boldsymbol{\mu}}_{t-1}, \boldsymbol{\lambda}_{t+1}(\boldsymbol{\sigma}_t, \hat{\boldsymbol{\mu}}_{t-1})) \\
&\quad - \boldsymbol{\sigma}_t^{\mathsf{T}} d(\hat{\boldsymbol{\mu}}_{t-1}, \boldsymbol{\lambda}_{t+1}(\boldsymbol{\sigma}_t, \hat{\boldsymbol{\mu}}_{t-1})) + A_{t-1}(\boldsymbol{\lambda}_{t+1}(\boldsymbol{\sigma}_t, \hat{\boldsymbol{\mu}}_{t-1})) \\
&\leq \sum_{s=1}^{t} \ell_s(u) + \boldsymbol{\sigma}_t^{\mathsf{T}} d(\hat{\boldsymbol{\mu}}_{t-1}, u) \\
&\quad - \boldsymbol{\sigma}_t^{\mathsf{T}} d(\hat{\boldsymbol{\mu}}_{t-1}, \boldsymbol{\lambda}_{t+1}(\boldsymbol{\sigma}_t, \hat{\boldsymbol{\mu}}_{t-1})) + A_{t-1}(\boldsymbol{\lambda}_{t+1}(\boldsymbol{\sigma}_t, \hat{\boldsymbol{\mu}}_{t-1})) \,.
\end{aligned}
$$

We obtain

$$
\begin{aligned}
A_t(u) - \boldsymbol{\sigma}_t^{\mathsf{T}} d(\hat{\boldsymbol{\mu}}_{t-1}, u) &= A_{t-1}(\boldsymbol{\lambda}_{t+1}(\boldsymbol{\sigma}_t, \hat{\boldsymbol{\mu}}_{t-1})) - \boldsymbol{\sigma}_t^{\mathsf{T}} d(\hat{\boldsymbol{\mu}}_{t-1}, \boldsymbol{\lambda}_{t+1}(\boldsymbol{\sigma}_t, \hat{\boldsymbol{\mu}}_{t-1})) \\
&= \sum_{s=1}^{t} \boldsymbol{\sigma}_{s-1}^{\mathsf{T}} d(\hat{\boldsymbol{\mu}}_{s-2}, \boldsymbol{\lambda}_{s+1}(\boldsymbol{\sigma}_s, \hat{\boldsymbol{\mu}}_{s-1})) - \boldsymbol{\sigma}_s^{\mathsf{T}} d(\hat{\boldsymbol{\mu}}_{s-1}, \boldsymbol{\lambda}_{s+1}(\boldsymbol{\sigma}_s, \hat{\boldsymbol{\mu}}_{s-1})) \,.
\end{aligned}
$$

End of the induction proof.

We now bound $A_t(u)$. First we write

$$
\begin{aligned}
A_t(u) - \boldsymbol{\sigma}_t^{\mathsf{T}} d(\hat{\boldsymbol{\mu}}_{t-1}, u) &= \sum_{s=1}^{t} \boldsymbol{\sigma}_{s-1}^{\mathsf{T}} d(\hat{\boldsymbol{\mu}}_{s-2}, \boldsymbol{\lambda}_{s+1}(\boldsymbol{\sigma}_s, \hat{\boldsymbol{\mu}}_{s-1})) - \boldsymbol{\sigma}_s^{\mathsf{T}} d(\hat{\boldsymbol{\mu}}_{s-1}, \boldsymbol{\lambda}_{s+1}(\boldsymbol{\sigma}_s, \hat{\boldsymbol{\mu}}_{s-1})) \\
&= \sum_{s=1}^{t} \boldsymbol{\sigma}_s^{\mathsf{T}} [d(\hat{\boldsymbol{\mu}}_{s-2}, \boldsymbol{\lambda}_{s+1}(\boldsymbol{\sigma}_s, \hat{\boldsymbol{\mu}}_{s-1})) - d(\hat{\boldsymbol{\mu}}_{s-1}, \boldsymbol{\lambda}_{s+1}(\boldsymbol{\sigma}_s, \hat{\boldsymbol{\mu}}_{s-1}))] \\
&\quad + \sum_{s=1}^{t} (\boldsymbol{\sigma}_{s-1} - \boldsymbol{\sigma}_s)^{\mathsf{T}} d(\hat{\boldsymbol{\mu}}_{s-2}, \boldsymbol{\lambda}_{s+1}(\boldsymbol{\sigma}_s, \hat{\boldsymbol{\mu}}_{s-1})) \,.
\end{aligned}
$$

We now bound separately the two sums. The first one uses the Lipschitz-continuity of $d$ and the fact that successive $\hat{\boldsymbol{\mu}}_t$ are not far from each other.

$$
\begin{aligned}
\mathbb{E} \sum_{s=1}^{t} \boldsymbol{\sigma}_s^{\mathsf{T}} &[d(\hat{\boldsymbol{\mu}}_{s-2}, \boldsymbol{\lambda}_{s+1}(\boldsymbol{\sigma}_s, \hat{\boldsymbol{\mu}}_{s-1})) - d(\hat{\boldsymbol{\mu}}_{s-1}, \boldsymbol{\lambda}_{s+1}(\boldsymbol{\sigma}_s, \hat{\boldsymbol{\mu}}_{s-1}))] \\
&= \mathbb{E} \sum_{s=1}^{t} \sigma_s^{k_{s-1}} [d(\hat{\mu}_{s-2}^{k_{s-1}}, \lambda_{s+1}^{k_{s-1}}(\boldsymbol{\sigma}_s, \hat{\boldsymbol{\mu}}_{s-1})) - d(\hat{\mu}_{s-1}^{k_{s-1}}, \lambda_{s+1}^{k_{s-1}}(\boldsymbol{\sigma}_s, \hat{\boldsymbol{\mu}}_{s-1}))] \\
&\leq \sum_{s=1}^{t} \mathbb{E}[\sigma_s^{k_{s-1}}] L |\hat{\mu}_{s-1}^{k_{s-1}} - \hat{\mu}_{s-2}^{k_{s-1}}| \leq CL \sum_{s=1}^{t} \mathbb{E}[\sigma_s^{k_{s-1}}] \frac{1}{N_{s-1}^{k_{s-1}}} \leq \frac{CL}{\eta} \sum_{s=1}^{t} \frac{\eta_s^{k_{s-1}}}{N_{s-1}^{k_{s-1}}} \,.
\end{aligned}
$$

For $\eta_t^k$ non-decreasing in $t$, $\boldsymbol{\sigma}_{s-1} - \boldsymbol{\sigma}_s$ has non-positive coordinates and the second sum is negative. We obtain

$$
\mathbb{E} \, A_t(u) \leq \mathbb{E} \, \boldsymbol{\sigma}_t^{\mathsf{T}} d(\hat{\boldsymbol{\mu}}_{t-1}, u) + \frac{CL}{\eta} \sum_{s=1}^{t} \frac{\eta_s^{k_{s-1}}}{N_{s-1}^{k_{s-1}}} \leq \frac{D \|\eta_t\|_1}{\eta} + \frac{CL}{\eta} \sum_{s=1}^{t} \frac{\eta_s^{k_{s-1}}}{N_{s-1}^{k_{s-1}}} \,.
$$

**First term of the regret.** Remark that $\boldsymbol{\lambda}_{t+1}(\boldsymbol{\sigma}, \hat{\boldsymbol{\mu}}_{t-1}) = \boldsymbol{\lambda}_t(\boldsymbol{\sigma} + e_{k_t}, \hat{\boldsymbol{\mu}}_{t-1})$. Let $f$ be the density of the distribution of $\sigma_t$. In expectation, the first term of the regret is

$$\mathbb{E}_{\boldsymbol{\sigma}_t}[\ell_t(\boldsymbol{\lambda}_t(\boldsymbol{\sigma}_t, \hat{\boldsymbol{\mu}}_{t-1})) - \ell_t(\boldsymbol{\lambda}_{t+1}(\boldsymbol{\sigma}_t, \hat{\boldsymbol{\mu}}_{t-1}))]$$

$$= \int_{\boldsymbol{\sigma}_t} [\ell_t(\boldsymbol{\lambda}_t(\boldsymbol{\sigma}_t, \hat{\boldsymbol{\mu}}_{t-1})) - \ell_t(\boldsymbol{\lambda}_t(\boldsymbol{\sigma}_t + e_{k_t}, \hat{\boldsymbol{\mu}}_{t-1}))] f(\boldsymbol{\sigma}_t) d\boldsymbol{\sigma}_t$$

$$= \int_{\boldsymbol{\sigma}_t} \ell_t(\boldsymbol{\lambda}_t(\boldsymbol{\sigma}_t, \hat{\boldsymbol{\mu}}_{t-1}))(f(\boldsymbol{\sigma}_t) - f(\boldsymbol{\sigma}_t - e_{k_t})) d\boldsymbol{\sigma}_t$$

By positivity of $\ell_t$ (since it is a divergence),

$$\mathbb{E}_{\boldsymbol{\sigma}_t}[\ell_t(\boldsymbol{\lambda}_t(\boldsymbol{\sigma}_t, \hat{\boldsymbol{\mu}}_{t-1})) - \ell_t(\boldsymbol{\lambda}_{t+1}(\boldsymbol{\sigma}_t, \hat{\boldsymbol{\mu}}_{t-1}))]$$

$$\leq \int_{\boldsymbol{\sigma}_t} \ell_t(\boldsymbol{\lambda}_t(\boldsymbol{\sigma}_t, \hat{\boldsymbol{\mu}}_{t-1})) \mathbb{I}\{f(\boldsymbol{\sigma}_t) - f(\boldsymbol{\sigma}_t - e_{k_t}) > 0\}(f(\boldsymbol{\sigma}_t) - f(\boldsymbol{\sigma}_t - e_{k_t})) d\boldsymbol{\sigma}_t$$

$$\leq D \int_{\boldsymbol{\sigma}_t} \mathbb{I}\{f(\boldsymbol{\sigma}_t) - f(\boldsymbol{\sigma}_t - e_{k_t}) > 0\}(f(\boldsymbol{\sigma}_t) - f(\boldsymbol{\sigma}_t - e_{k_t})) d\boldsymbol{\sigma}_t$$

$$\leq D \int_{\boldsymbol{\sigma}_t} \mathbb{I}\{f(\boldsymbol{\sigma}_t) - f(\boldsymbol{\sigma}_t - e_{k_t}) > 0\} f(\boldsymbol{\sigma}_t) d\boldsymbol{\sigma}_t$$

$$= D \int_{\sigma_t^{k_t} \leq 1} f(\boldsymbol{\sigma}_t) d\boldsymbol{\sigma}_t$$

$$= D(1 - e^{-\eta/\eta_t^{k_t}})$$

$$\leq D\eta/\eta_t^{k_t} .$$

**Putting things together.** Choose $\eta_t^k = \sqrt{N_{t-1}^k}$.

$$\mathbb{E}\, R_t \leq D \frac{\|\eta_t\|_1}{\eta} + \frac{CL}{\eta} \sum_{s=1}^{t} \frac{\eta_s^{k_{s-1}}}{N_{s-1}^{k_{s-1}}} + D\eta \sum_{s=1}^{t} \frac{1}{\eta_s^{k_s}} \leq \sqrt{Kt} \left( \frac{D + 2CL}{\eta} + 2D\eta \right) .$$

$\square$

**Approximation of $q_t$ by an empirical distribution.** We want the $k$-player to use optimistic best-response to $\boldsymbol{q}_t$. This requires the computation of

$$\operatorname*{argmax}_{k \in [K]} U_t^k \qquad \text{with } U_t^k = \max_{\xi \in \{a_t^k, b_t^k\}} \mathbb{E}_{\boldsymbol{\lambda} \sim \boldsymbol{q}_t} d(\xi, \lambda^k) .$$

for some values $a_t^k, b_t^k$.

Since we cannot compute an expectation under $\boldsymbol{q}_t$ exactly, we compute instead the expectation under an empirical distribution based on $t$ samples $\boldsymbol{\lambda}_t^{(1)}, \ldots, \boldsymbol{\lambda}_t^{(t)}$ of $\boldsymbol{q}_t$. For all $\xi$, $d(\xi, \lambda^k)$ is bounded by $D$. Hence, by Hoeffding's inequality,

$$\mathbb{P}\left\{ \frac{1}{t} \sum_{j=1}^{t} d(\xi, \lambda_t^{(j)k}) - \mathbb{E}_{\boldsymbol{\lambda} \sim \boldsymbol{q}_t} d(\xi, \lambda^k) \geq \sqrt{\frac{3D^2 \log(t)}{2t}} \right\} \leq \frac{1}{t^3} .$$

In the concentration analysis of the algorithm, we replace $\mathcal{E}_t$ by $\mathcal{E}_t \cap \mathcal{E}'_t$ with

$$\mathcal{E}'_t = \left\{ \forall k \in [K], \forall s \leq t, \forall \xi \in \{a_s^k, b_s^k\} \frac{1}{s} \sum_{j=1}^{s} d(\xi, \lambda_s^{(j)k}) - \mathbb{E}_{\boldsymbol{\lambda} \sim \boldsymbol{q}_s} d(\xi, \lambda^k) \leq D\sqrt{\frac{3 \log(t)}{2t}} \right\}$$

It verifies $\sum_{t=1}^{+\infty} \mathbb{P}(\mathcal{E}'^c_t) \leq 2K \sum_{t=1}^{+\infty} 1/t^2 \leq K\pi^2/3$.

Under the event $\mathcal{E}'_t$,

$$\sum_{s=1}^{t} \frac{1}{s} \sum_{j=1}^{s} d(\xi, \lambda_s^{(j)k}) - \sum_{s=1}^{t} \mathbb{E}_{\boldsymbol{\lambda} \sim \boldsymbol{q}_s} d(\xi, \lambda^k) \leq D\sqrt{\frac{3}{2} t \log(t)} .$$

We obtain that the procedure based on these empirical distributions has $\mathcal{O}(\sqrt{t \log t})$ regret.

# F  On the statistical assumptions

## F.1  The sub-Gaussian assumption

The natural coordinate-wise concentration events for exponential families have the form $N_t^k d(\hat{\mu}_t^k, \mu^k) \leq c$ for some constant $c > 0$. In our proofs, we need then to relate $d(\hat{\mu}_t^k, \lambda^k)$ and $d(\mu^k, \lambda^k)$ for a given $\lambda^k$ under such a concentration constraint. However, we now show that for some convex function $\phi$ (such that $d$ is the associated Bregman divergence), these two quantities could be very far apart even under the constraint $d(\hat{\mu}_t^k, \mu^k) = 0$.

If $d(\hat{\mu}_t^k, \mu^k) = 0$, we have the equalities

$$d(\hat{\mu}_t^k, \lambda^k) - d(\mu^k, \lambda^k) = d(\hat{\mu}_t^k, \mu^k) + (\hat{\mu}_t^k - \mu^k)(\phi'(\mu^k) - \phi'(\lambda^k))$$
$$= (\hat{\mu}_t^k - \mu^k)(\phi'(\mu^k) - \phi'(\lambda^k)) \,.$$

Let $\phi : \mathbb{R} \to \mathbb{R}$ be defined by $\phi(x) = \max\{0, x\}$. Let $\lambda^k = 1$, $\mu^k = -1$ and $\hat{\mu}_t^k < -1$. Then

$$d(\hat{\mu}_t^k, \mu^k) = 0 \,,$$
$$d(\hat{\mu}_t^k, \lambda^k) - d(\mu^k, \lambda^k) = |\hat{\mu}_t^k - \mu^k| \,.$$

In that example, the constraint on $d(\hat{\mu}_t^k, \mu^k)$ is not sufficient to bound $d(\hat{\mu}_t^k, \lambda^k) - d(\mu^k, \lambda^k)$.

The example exploits the piecewise linearity of $\phi$. Such a function $\phi$ cannot arise from an exponential family. Indeed, for an exponential family $\phi$ is the convex conjugate of a cumulant generating function. In particular, $\phi$ is strictly convex. But it could still have very low curvature (for example for an exponential distribution with high mean). The sub-Gaussian assumption ensures that $\phi$ is strongly convex.

Our work and previous parametric pure exploration papers treat $d$ as a general Bregman divergence. The present example shows that either we need to also use more specific properties of $d$ due to the fact that it is a Kullback-Leibler divergence, or we need to impose additional assumptions like sub-Gaussianity.

## F.2  The upper bound assumption

A first way to relax the assumption that $\mathcal{M} \subseteq [\mu_{\min}, \mu_{\max}]^K$ is to remark that we do not need to bound $d(\mu, \lambda)$ for any $\mu$ and $\lambda$.

For $\boldsymbol{\mu} \in \mathcal{M}$ and $\boldsymbol{w} \in \triangle_K$, let $\boldsymbol{\lambda}(\boldsymbol{\mu}, \boldsymbol{w}) = \text{argmin}_{\boldsymbol{\lambda} \in \neg i} \sum_{k=1}^{K} w^k d(\mu^k, \lambda^k)$. Our proofs are valid for example under the following assumption.

**Assumption 4.** There exists $D > 0$ and $L > 0$ such that for all $\boldsymbol{w} \in \triangle_K$, for all $\boldsymbol{\mu} \in \mathcal{M}$, $\|d(\boldsymbol{\mu}, \boldsymbol{\lambda}(\boldsymbol{\mu}, \boldsymbol{w}))\|_\infty \leq D$ and $\|\phi'(\boldsymbol{\mu}) - \phi'(\boldsymbol{\lambda}(\boldsymbol{\mu}, \boldsymbol{w}))\|_\infty \leq L$.

We could also use the concentration events to replace it with weaker hypotheses. Under event $\mathcal{E}_t$ and with Assumption 1, for all $s \leq t$, $\|d(\hat{\boldsymbol{\mu}}_s, \boldsymbol{\mu})\|_\infty \leq f(t)$ and $\|\hat{\boldsymbol{\mu}}_s - \boldsymbol{\mu}\|_\infty \leq \sqrt{2\sigma^2 f(t)}$. That is, we get from concentration only, without assumptions, that $\hat{\boldsymbol{\mu}}_t$ is in a bounded set around $\boldsymbol{\mu}$. We can then quantify $L$ and $D$ on that set.

Let $L_{\boldsymbol{\mu}} = \sup_{\boldsymbol{w} \in \triangle_k} \max_k |\phi'(\mu^k) - \phi'(\lambda(\boldsymbol{\mu}, \boldsymbol{w})^k)|$.

**Assumption 5.** For all $\boldsymbol{\mu} \in \mathcal{M}$, $L_{\boldsymbol{\mu}}$ is finite.

This is true for BAI, where $L_{\boldsymbol{\mu}} \leq \phi'(\max_k \mu^k) - \phi'(\min_k \mu^k)$.

**Assumption 6.** There exists $M > 0$ such that $\boldsymbol{\mu} \mapsto L_{\boldsymbol{\mu}}$ is $M$-Lipschitz for the $\ell^\infty$ norm.

This is true for BAI on sets on which $\phi'$ is Lipschitz. For example, it is true on $\mathbb{R}$ for Gaussian arm distributions, but is still only true in intervals of the form $[\varepsilon, 1 - \varepsilon]$ for Bernoulli distributions.

Then for any $\boldsymbol{\mu}, \boldsymbol{\xi}$ and $\boldsymbol{\lambda_\mu}$ minimal point for $\boldsymbol{\mu}$, for any coordinate $k \in [K]$ (omitted in the computations),

$$
\begin{aligned}
d(\mu, \lambda_\mu) &= d(\xi, \lambda_\mu) + (\mu - \xi)(\phi'(\mu) - \phi'(\lambda_\mu)) - d(\xi, \mu) \\
&\geq d(\xi, \lambda_\mu) - |\mu - \xi| L_\mu - d(\xi, \mu) \\
&\geq d(\xi, \lambda_\mu) - |\mu - \xi| L_\xi - M(\mu - \xi)^2 - d(\xi, \mu) \\
&\geq d(\xi, \lambda_\mu) - L_\xi \sqrt{2\sigma^2 \min\{d(\mu, \xi), d(\xi, \mu)\}} - 2\sigma^2 M \min\{d(\mu, \xi), d(\xi, \mu)\} - d(\xi, \mu)
\end{aligned}
$$

Examples for the quantities used in the proofs:

$$
d(\mu, \lambda_\mu) \geq d(\hat\mu_{s-1}, \lambda_\mu) - L_\mu \sqrt{2\sigma^2 d(\hat\mu_{s-1}, \mu)} - d(\hat\mu_{s-1}, \mu)
$$
$$
d(\hat\mu_t, \lambda_{\hat\mu_t}) \geq d(\mu, \lambda_{\hat\mu_t}) - L_\mu \sqrt{2\sigma^2 d(\hat\mu_t, \mu)} - 2\sigma^2 M d(\hat\mu_t, \mu) - d(\mu, \hat\mu_t)
$$

The proofs must then be adapted to account for the additional terms in these inequalities.

# G   Numerical Experiments

## G.1   Best Arm

(a) Bernoulli bandit $\boldsymbol{\mu} = (0.5, 0.45, 0.43, 0.4)$, $\boldsymbol{w}^* = (0.42, 0.39, 0.14, 0.06)$

(b) Bernoulli bandit $\boldsymbol{\mu} = (0.3, 0.21, 0.2, 0.19, 0.18)$, $\boldsymbol{w}^* = (0.34, 0.25, 0.18, 0.13, 0.10)$

Figure 2: Best Arm experiments from [13]. In both cases $\delta = 0.1$. Plots show 3000 runs.

(a) $\delta = 0.1$

(b) $\delta = 0.01$

Figure 3: Best Arm experiment from [24]. Gaussian bandit $\boldsymbol{\mu} = (1., 0.85, 0.8, 0.7)$, $\boldsymbol{w}^* = (0.41, 0.38, 0.15, 0.06)$. Plots show 3000 runs.

## G.2  Minimum Threshold

(a) Gaussian bandit $\boldsymbol{\mu} = (-1, \ldots, 1)$ with $K = 10$ arms and $\delta = e^{-23}$, $\boldsymbol{w}^* = \boldsymbol{e}_1$

(b) Gaussian bandit $\boldsymbol{\mu} = (0.5, \ldots, 1)$ with $K = 5$ arms and $\delta = e^{-7}$, $\boldsymbol{w}^* = (0.38, 0.24, 0.17, 0.12, 0.09)$

Figure 4: Minimum Threshold experiments from [20] with threshold $\gamma = 0$. Plots show 5000 runs.

(a) $\delta = 0.1$

(b) $\delta = 0.0001$

(c) $\delta = 10^{-10}$

(d) $\delta = 10^{-20}$

Figure 5: Minimum Threshold experiment (new): Gaussian bandit $\boldsymbol{\mu} = (0.5, 0.6)$ with threshold $\gamma = 0.6$, $\boldsymbol{w}^* = \boldsymbol{e}_1$. Note the excessive sample complexity of Track-and-Stop (T-C and T-D). Plots show 5000 runs.

The reason for the bad performance of Track-and-Stop in Figure 5 is that with small but non-negligible probability the algorithm finds $\hat{\mu}_t^1 \gg \gamma$ estimated too high at some early $t$. In this situation $\boldsymbol{w}^*(\boldsymbol{\mu}_t)$ will be $\boldsymbol{e}_2$ (exactly if $\hat{\mu}_t^2 \leq \gamma$, approximately if $\hat{\mu}_t^2 > \gamma$), and constantly pulling arm 2 will not correct the estimate of arm 1. **T** relies on forced exploration to correct the estimate.