[Reviews · NeurIPS 2019]

Reviewer 1



The paper is easy to read. The authors convey their main ideas clearly. The main issue with this paper is that lower bounds are hard to interpret. They do show clearly how these lower bounds match with other results in the top-k bandits or the best-arm identification problem. The regret bounds are also linearly dependent on the number of arm K. This poses problems when the number of arms is very big and therefor this analysis only seems to be interesting in the case when K << T. Therefore while these are finite time bounds, the time still has to significantly bigger than the number of arms. Typo in line 4 of algorithm 1 in the subscript of the argmin.

Reviewer 2



This paper adds a link between the principle of optimism in the face of uncertainty (widely used in regret minimization setting) and pure exploration problems, and proposes algorithms based on this principle. This idea is natural, and it is not surprised that the proposed algorithms can achieve non-asymptotic guarantee. The paper is clearly written, and it is technically sound. The analysis is a combination of known techniques, mainly from the bandit literature and reference [9], which makes the paper not that novel. The paper proposes several algorithms, but the numerical experiment is not comprehensive.

Reviewer 3



(1) This work studies the complexity of pure exploration when the confidence level is a constant. The approach is to solve the minimax optimization problem in the lower bound by viewing it as a two-player game and defining the players' learning dynamics. The authors also discuss a possible trade-off between the power of optimization oracles and the sample complexity guarantee. This seems to me a novel contribution to the field. (2) Overall, the paper is well-written and not too difficult to follow. I was able to follow the proof sketches in Section 3 and the technical claims in the paper seem valid to me. (3) My concern about this work is about the significance of the contributions, due to a lack of comparison to previous approaches and results; see Part 5 for details. *** added after author feedback *** After reading the author feedback, I thank the authors for addressing my concerns and would encourage the authors to add the discussion to the final version.

[Author Response · NeurIPS 2019]

We thank the reviewers for all their remarks and comments. We address these remarks separately below.

**Reviewer 1:** Our bounds indeed require the horizon $T$ to be bigger than the number of arms $K$. We study an
intermediate confidence regime which is non-asymptotic but in which the algorithm still gets to access several samples
per arm. The setting where the algorithm should not even pull each arm once (due to a strong structure and low
confidence level) is a challenge that we do not address.

You mention that the lower bounds we use are difficult to compare to the bounds obtained in other works and we agree
that their non-explicit nature can have that effect. For best-arm with Gaussian arms with variance $\sigma^2$, it is shown in [13]
that the lower bound $T^*$ such that $\liminf_{\delta \to 0} \frac{\mathbb{E}[\tau_\delta]}{\log(1/\delta)} \geq T^*$ verifies $\sum_{k \neq *} \frac{2\sigma^2}{\Delta_k^2} \leq T^* \leq 2 \sum_{k \neq *} \frac{2\sigma^2}{\Delta_k^2}$. On the upper
bound side, we can divide the existing work in two categories: papers that get only asymptotic optimality, like [13,
20, 24]; and papers that get finite confidence guarantees but not asymptotically optimal (with the right multiplicative
factors), like [5, 16]. In the second case, more effort is spent towards making the other terms of the bound small. For
the Gaussian top-k arms problem, the difference between the lower bound we use and the gap-based one found for
example in [Chen et al., Nearly Instance Optimal Sample Complexity Bounds for Top-k Arm Selection, 2017] is also a
multiplicative factor. For non-Gaussian arms, the difference can be much larger. See [13, section 2.2] for a discussion.

**Reviewer 2:** We disagree with your assessment of the contributions of the paper, expressed in that sentence: "This
paper adds a link between the principle of optimism in the face of uncertainty (widely used in regret minimization
setting) and pure exploration problems, and proposes algorithms based on this principle." Our design exploits a game
point of view through a combination of two adversarial algorithms playing against each other, one of which deals with
uncertainty by using optimism. The optimism is more akin to a trick than the main feature. The main contribution is
to show how the interaction of the two players iteratively solve the lower bound problem to tend towards the optimal
sampling behaviour (and complexity). No other stochastic bandit paper uses a similar design. The reference [9] notes
that the lower bound can be seen as a game and uses that observation to derive lower bounds, but that paper does not
use this hindsight to form algorithms (their algorithm is a slightly modified track-and-stop).

For the experimental part, we are inclined towards keeping it fairly light in the main part of the paper due to space
constraints. We can however add comparisons to more algorithms in the appendix. As our algorithm can deal with
any exploration problem on which one of the required oracles can be implemented, and not only best-arm problems,
we compared to similar general algorithms. We will include the algorithm of https://arxiv.org/abs/1602.08448 in the
experiments of appendix G.

We agree that the computational complexity claims would be better supported with data, which we will include. To give
a quick idea: on best-arm identification (Figure 1a, section 4), taking the time per iteration of uniform sampling as
1, the algorithms D, M, T and O have iteration times of 6.4, 3.8, 121 and 526 respectively. Our new algorithm D has
same order of computational complexity as M [24] up to an overhead due to the computation of the optimism terms.
Track-and-Stop (T) is slower and our new algorithm O is again slower, as predicted (it uses a more complicated oracle).
On the thresholding task, all oracles are closed form and all algorithms have similar iteration times (6.1, 5.1, 7.1, 5.2).

**Reviewer 3:** We doubt that a meaningful moderate confidence bound can be derived from [13], for reasonable
confidence $\delta$. Their continuity based argument introduces a quantity $\varepsilon > 0$ and the forced exploration leads to a bound
that depends on $\varepsilon$ and the parameters of the problem through polynomial terms with high exponents (for example $1/\varepsilon^4$).
$\varepsilon$ must then be chosen as a function of $\delta$ to obtain a moderate confidence bound. Our best efforts lead to a bound with a
lower order term proportional to $(\log(1/\delta))^{11/12}$, which is $o(\log(1/\delta))$ but potentially still big for moderate confidence.

We evaluated track-and-stop (TaS) experimentally in the paper, and the experiments (in particular Figure 1b and Figures 5) show that while TaS is asymptotically optimal, the empirical performance may be poor, even for $\delta = e^{-20}$. On the left, we reproduced Figure 1b with larger markers for the mean (crosses). TaS (called T-C and T-D) performs much worse that even uniform sampling.

About [Chen et al., 2017]: we indeed should have discussed it since it addresses the same problem. We now discuss it and will include this comparison in the paper. Their algorithm has several features in common with TaS: a phase of forced exploration restricts the candidates answers to only 1 (through the use of a confidence region) and forms estimates of the parameters; the algorithm then verifies that it is indeed the correct answer up to the required confidence level by using a plug-in estimate of the solution to the lower bound problem.

52 Their definition of "optimal" is not the same as ours. Our algorithm verifies $\lim_{\delta \to 0} \mathbb{E}[\tau_\delta]/\log(1/\delta) = T^*$, where $T^*$
53 is the lower bound complexity. They show a $256T^*$ bound. Their algorithm's complexity is optimal in the sense that it
54 is proportional to $T^*$. Ours matches the lower bound with the right multiplicative constant.

[Meta-Review · NeurIPS 2019]

The reviewers liked this paper, and so did I. Nice work. A small remark: perhaps the paper by Chan and Lai "Sequential generalized likelihood ratios and adaptive treatment allocation for optimal sequential selection" should be given some love as a precursor to the Track-and-Stop algorithm of Kaufmann et al.